# CURS : an exact method for sampling on Riemannian manifolds

**Isabella Costa Maia**                                   *isabella.costa-maia@grenoble-inp.fr*
*GIPSA-lab, University Grenoble Alpes, CNRS, Grenoble-INP*

**Marco Congedo**                                    *marco.congedo@gipsa-lab.grenoble-inp.fr*
*GIPSA-lab, University Grenoble Alpes, CNRS, Grenoble-INP*

**Pedro L. C. Rodrigues**                              *pedro.rodrigues@univ-grenoble-alpes.fr*
*Univ. Grenoble Alpes, Inria, CNRS, Grenoble INP, LJK*

**Salem Said**                                       *salem.said@univ-grenoble-alpes.fr*
*Univ. Grenoble Alpes, CNRS, Grenoble INP, LJK*

**Reviewed on OpenReview:** *https://openreview.net/forum?id=LY9ecALVDm*

## Abstract

The present work introduces curvature-based rejection sampling (CURS). This is a method for sampling from a general class of probability densities defined on Riemannian manifolds. It can be used to sample from any probability density which "depends only on distance". The idea is to combine the statistical principle of rejection sampling with the geometric principle of volume comparison. CURS is an exact sampling method and (assuming the underlying Riemannian manifold satisfies certain technical assumptions) it has a particularly moderate computational cost. The aim of the present work is to show that there are many applications where CURS should be the user's method of choice for dealing with relatively low-dimensional scenarios.

## 1 Introduction

The present work introduces a new method for sampling from probability densities defined on Riemannian manifolds. This method is called curvature-based rejection sampling (CURS). It addresses the problem of sampling from a probability density function which "depends only on distance". Specifically, let $M$ be a Riemannian manifold and let $x_o$ be some point in $M$. CURS can be used to sample from any probability density $p$ on $M$, of the form

$$p(x) \propto f(r(x)) \text{ where } r(x) = d(x_o, x) \tag{1}$$

Here, $p \propto f$ means $p$ is proportional to $f$, and $d(x_o, x)$ denotes the Riemannian distance between $x_o$ and $x$ in $M$ (the precise definition of a probability density on $M$ is given in the following Section 2).

CURS is a geometric rejection sampling method which has the following advantages :

- it is a rather general method, since it applies to any density $p$ of the form (1) and to a large class of manifolds $M$. For example, it applies successfully whenever $M$ is a so-called Hadamard manifold, such as a manifold of covariance matrices (real, complex, Toeplitz, *etc.*) (Lee, 2018; Said et al., 2018), but is by no means limited to this special case of Hadamard manifolds.

- it is an exact method, which produces samples that immediately follow the density $p$, with no need for any kind of burn-in period. This is in contrast to existing Markov-Chain Monte Carlo methods.

- the computational cost of a single iteration of this method is quite moderate.

- it is often possible to theoretically evaluate the time complexity of this method, by computing its rejection probability. In this way, one has a theoretical performance guarantee.

The introduction of CURS was motivated by the specific problem of sampling from a Gaussian density

$$p(x) \propto \exp\left[-\frac{d^2(x_o, x)}{2\sigma^2}\right] \text{ where } \sigma > 0 \qquad (2)$$

which is defined on the manifold $M$ of real covariance matrices. In the notation of (1), this corresponds to $f(r) = \exp(-r^2/2\sigma^2)$. Section 5, below, will return to this problem.

At present, existing methods for sampling on manifolds can be divided into exact and approximate methods. Exact methods tend to be specialized, applicable only to specific probability distributions and specific classes of manifolds. For example, there exist highly effective methods for sampling from uniform distributions on compact Lie groups, or from classical directional distributions on Stiefel and Grassmann manifolds (Meckes, 2019; Chikuse, 2003). Clearly, the major drawback of such methods is that they are too specialized, as they are of no immediate use beyond their originally intended goal.

Approximate methods are based on geometric Markov Chain Monte Carlo techniques (geometric MCMC). They are very general, as they can be used to sample from any positive density, on any Riemannian manifold which satisfies rather mild technical conditions. Famous examples of such methods include Hamiltonian Monte Carlo and geometric Langevin MCMC (Girolami & Calderhead, 2011; Bharath et al., 2025). Their main drawback is their computational complexity, due to the use of often complicated discretization schemes.

CURS follows a kind of middle ground which allows it to make the best of both worlds. Being neither too specialized nor completely general, it is able to perform exact sampling at a moderate computational cost. Its main drawback is that, being a rejection sampling method, its performance is significantly affected by the curse of dimensionality.

Still, users should find it helpful to know about CURS, and retain it as part of their toolbox, due to its highly advantageous performance-complexity tradeoff in relatively low-dimensional scenarios. For example, these may arise in the context of manifold-valued latent-variable modelling, where the latent manifolds are typically of lower dimension, but can have pretty general geometries (Skopek et al., 2020). Existing models, such as the hyperspherical or hyperbolic variational autoencoders (Davidson et al., 2018; Mathieu et al., 2019), only deal with elementary geometries, and use sampling methods that mostly apply to pre-determined, specific families of distrbutions.

CURS may also be useful as a part of other sampling methods, such as the Riemannian proximal sampler (Guan et al., 2025). This is a high-accuracy sampling method, which can produce nearly-exact independent samples from target densities that satisfy a logarithmic Sobolev inequality. A single iteration of a Riemannian proximal sampler involves two rejection sampling schemes, each with a proposal density of the form (1). Potentially, CURS can improve both the performance and the complexity of these rejection sampling routines, and therefore the overall quality of the proximal sampler. This is greatly helpful since the applicability of the proximal sampler well excedes that of CURS, which is limited to target densities of the above form (1).

Thus, either directly or indirectly (as a component of other methods), CURS should be a helpful contribution to sampling on manifolds and to its applications, which include generative modelling, Bayesian learning and optimisation, and also statistical physics (Durmus & Moulines, 2017; Patterson & Teh, 2013; Leimkuhler & Matthews, 2016; De Bortoli et al., 2022).

The present work is organised as follows. Section 2 explains the general idea behind the CURS method. Section 3 formalises this idea into pseudo-code instructions. Section 4 discusses the geometric interpretation of CURS, while Section 5 applies it to the problem of sampling from a Gaussian density of the form (2). Finally, Section 6 is concerned with a more general version of CURS, extending its applicability to a larger class of Riemannian manifolds. Code for reproducing the experiments from Section 5 is available at `https://github.com/isacostamaia/CURS-An-exact-method-for-sampling-on-Riemannian-manifolds/`.

## 2 The idea behind CURS

### 2.1 Geodesic spherical coordinates

Consider the problem of sampling from a probability density $p$ of the form (1). Since $p(x)$ depends only on the distance $r(x)$, one may hope to perform a separation of variables, by introducing geodesic spherical coordinates (Chavel, 2006; Lee, 2018). These coordinates parameterise the point $x \in M$ using two quantities, the distance $r$ and the direction $s$ (in fact, $s$ is a unit-length tangent vector to $M$ at the point $x_o$). Then, $x$ is obtained by moving for a distance $r$ along a geodesic curve starting at $x_o$ and whose initial velocity is equal to the vector $s$.

In terms of the Riemannian exponential mapping (Chavel, 2006; Lee, 2018)

$$x = \mathrm{Exp}_{x_o}(r\,s) \tag{3}$$

which will be abbreviated to $x = x(r, s)$.

In general, geodesic spherical coordinates do not regularly cover the entire manifold $M$, but are limited to a subset $D_{x_o}$ called the domain of injectivity. For now (to simplify the presentation) the following assumption is made.

**Assumption A** $D_{x_o} = M$, so the mapping $(r, s) \mapsto x(r, s)$ in (3) is a diffeomorphism from $(0, \infty) \times S_{x_o}M$ to $M - \{x_o\}$ (here, $S_{x_o}M$ denotes the set of unit-length tangent vectors to $M$ at $x_o$).

This assumption is indeed verified when $M$ is a Hadamard manifold, such as a manifold of covariance matrices (real, complex, Toeplitz, *etc.*) (Lee, 2018; Said et al., 2018). It will be dropped later on, in Section 6.

One hopes to sample $x$ by first separately sampling $r$ and $s$ and then replacing them into (3). The major stumbling block for this approach will be the difficulty in expressing the Riemannian volume measure of $M$. To say that $p$ is a probability density on $M$ means that the probability distribution of a random point $x$ (when $x$ is sampled from $p$) is given by

$$P(dx) = p(x)\,\mathrm{vol}(dx) \tag{4}$$

where vol denotes the Riemannian volume measure of $M$. From (1), the joint distribution of $r$ and $s$ is then

$$P(dr \times ds) \propto f(r)\,\mathrm{vol}(dr \times ds) \tag{5}$$

where $\mathrm{vol}(dr \times ds)$ is just $\mathrm{vol}(dx)$, but expressed in terms of $r$ and $s$. This is given as follows.

Note that $S_{x_o}M$ is the unit sphere in the tangent space to $M$ at $x_o$. Let $\omega$ denote the surface area measure on this sphere. Then, the following formula holds (Chavel, 2006)

$$\mathrm{vol}(dr \times ds) = |\det A(r, s)|\,dr\omega(ds) \tag{6}$$

where the matrix $A(r, s)$ is obtained by solving a second-order linear differential equation, known as the matrix Jacobi equation (Chavel, 2006) (Page 114). Replacing (6) into (5) yields

$$P(dr \times ds) \propto f(r)\,|\det A(r, s)|\,dr\omega(ds) \tag{7}$$

Thus, in order to sample $r$ and $s$, one must sample from the joint distribution (7), and this is where the difficulties come in. Since $|\det A(r, s)|$ depends on both $r$ and $s$, these two cannot be sampled separately (in other words, independently). Rejection sampling comes to the rescue, as seen in the following subsection.

### 2.2 Introducing rejection sampling

Looking back to (7), the following observation can be made. In order to sample $r$ and $s$ independently, one should aim to have

$$|\det A(r, s)| = \text{ a function of } r \ \times \ \text{ a function of } s$$

Unfortunately, such a factorisation holds only quite rarely. However, in many situations, one can still replace the equality by an inequality. This is the case whenever the following (very mild) assumption holds.

**Assumption B** the sectional curvatures of $M$ are bounded below by some negative number $-\kappa^2$.

Under this assumption, Bishop's volume comparison theorem states that the following inequality holds (Chavel, 2006) (Page 130),

$$|\det A(r,s)| \leq \left(\kappa^{-1}\sinh(\kappa r)\right)^{d-1} \tag{8}$$

where $d$ is the dimension of $M$. Now, the right-hand side is indeed the product of a function of $r$ by a function of $s$ (the latter being a constant function, equal to 1).

Assumption B (lower bound on sectional curvatures), and therefore inequality (8), is verified in many, and even most, realistic situations. In particular, it holds for spaces of real or complex covariance matrices, with $\kappa = 1/\sqrt{2}$ (see Appendix B).

With (8) at hand, it becomes possible to apply a rejection sampling strategy, along the general lines explained in (Robert & Casella, 2004). This leads to the following formulation of the CURS method.

The target joint density of $r$ and $s$ in (7) reads

$$p(r,s) = \frac{1}{Z} \times f(r)\,|\det A(r,s)| \tag{9}$$

where $Z$ is a normalising constant (the density is with respect to the reference measure $dr\omega(ds)$). Consider the proposal joint density (of course, with respect to the same reference measure)

$$g(r,s) = \frac{1}{Z_\kappa} \times f(r)\left(\kappa^{-1}\sinh(\kappa r)\right)^{d-1} \tag{10}$$

where $Z_\kappa$ is again a normalising constant. Then, note from (8)

$$p(r,s) \leq T \times g(r,s) \text{ where } T = \frac{Z_\kappa}{Z} \tag{11}$$

This is the key inequality required for rejection sampling. According to (Robert & Casella, 2004), to sample from $p(r,s)$, it is enough to sample from the proposal $g(r,s)$ and then reject any samples which do not verify

$$U \leq \frac{1}{T} \times \frac{p(r,s)}{g(r,s)} \tag{12}$$

where $U$ is an independently generated random variable, with uniform distribution in the unit interval $[0,1]$. As explained in the previous Section 2, samples $x$ from the original density $p(x)$ are then obtained by replacing $r$ and $s$ into (3).

CURS is just a direct implementation of this idea. Everything works out because the proposal density $g(r,s)$ does not depend on $s$. This means that $r$ and $s$ can be sampled independently under this density. Moreover, it implies that $s$ follows a uniform distribution with respect to the surface area measure $\omega$ on the sphere $S_{x_o}M$. This makes it particularly easy to sample $s$.

## 3 CURS in pseudo-code

It is now possible to formulate the CURS method in pseudo-code. Its major steps turn out to be the following.

A few comments are in order.

For step 1, sampling a uniform distribution on a sphere is a well-known procedure. First, $s$ is sampled from a standard normal distribution in the tangent space to $M$ at $x_o$. Then, $s$ is replaced with $s/\|s\|$ which clearly belongs to the unit sphere $S_{x_o}M$ ($\|s\|$ denotes the norm of $s$ with respect to the Riemannian metric of $M$). For this step, the vector $s$ should always be expressed in an orthonormal basis of the tangent space at $x_o$.

For step 2, if $f(r)$ is log-concave, then the density $g(r)$ in (13) will also be log-concave. It can then be efficiently sampled using the universal black-box algorithm of (Devroye, 1986).

---

**Algorithm 1** CURS algorithm

---

    ## Sample $(r, s)$ from $g(r, s)$ in (10)
1: Sample $s$ uniformly on the sphere $S_{x_o}M$
2: Sample $r > 0$ from density

$$g(r) \propto f(r) \left(\kappa^{-1} \sinh(\kappa r)\right)^{d-1} \tag{13}$$

    ## Apply the rejection procedure
3: Sample $U \sim \mathcal{U}(0, 1)$.
4: Evaluate

$$U > \frac{|\det A(r, s)|}{\left(\kappa^{-1} \sinh(\kappa r)\right)^{d-1}} \tag{14}$$

5: **if** condition (14) holds **then**
6:     Reject $(r, s)$ and return to step 1.
7: **else**
8:     Accept $(r, s)$ and put $x = x(r, s)$ as in (3).

---

For step 4, note that the condition in (14) is equivalent to (12), according to the definition of $T$ in (11). This step requires a single evaluation of $\det A(r, s)$. It is here assumed that this is available.

**Assumption C** $\det A(r, s)$ is known explicitly, and can be readily evaluated for any $(r, s) \in (0, \infty) \times S_{x_o}M$. Oftentimes, this is indeed true (see Section 5). Eventually, if this is not the case, accurate numerical evaluation of $A(r, s)$ remains a straightforward option. This is not discussed here, since it is felt that situations where $\det A(r, s)$ is known in closed form already cover an extensive range of applications.

The main indicator of the time complexity of CURS is the acceptance probability

$$\Pi = \frac{1}{T} = \frac{Z}{Z_\kappa} \tag{15}$$

In fact, $T$ is the expected number of iterations necessary to produce a single new sample. In order to have a theoretical performance guarantee for CURS, one should be capable of computing $\Pi$. While $Z_\kappa$ essentially reduces to a one-dimensional integral, the target density normalising constant $Z$ will be a multiple integral. The following expressions are completely general (see Appendix A)

$$Z = \int_{S_{x_o}M} \int_0^\infty f(r) \left|\det A(r, s)\right| dr \omega(ds) \tag{16}$$

$$Z_\kappa = \Omega_{d-1} \int_0^\infty f(r) \left(\kappa^{-1} \sinh(\kappa r)\right)^{d-1} dr \tag{17}$$

where $\Omega_{d-1}$ is the surface area of a $(d-1)$-dimensional sphere (in other words, of $S_{x_o}M$). Clearly, the core problem related to computing $\Pi$ is the evaluation of $Z$.

## 4 Geometric interpretation

CURS is born from the marriage of the statistical principle of rejection sampling to the geometric principle of volume comparison. This confers a new geometric meaning onto the proposal density (10) and the rejection condition (14). In order to uncover this new meaning, recall Bishop's volume comparison inequality (8). This inequality becomes an equality in exactly one case: when $M$ is a hyperbolic space, a space of constant negative curvature equal to $-\kappa^2$. If $M$ were in fact such a hyperbolic space, the target joint density (9) would be identical to the proposal density (10). In other words, the proposal density pretends that the space $M$ is a hyperbolic space with constant curvature $-\kappa^2$. The role of the rejection condition (14) is then to recover the true geometry of $M$.

The main difference between the hyperbolic "proposal geometry" and the true geometry of $M$ comes from anisotropy. In the hyperbolic case, a geodesic starting from $x_o$ in any initial direction $s$ only meets one value of the sectional curvature, $-\kappa^2$. In the general case, different directions $s$ can meet different values of the sectional curvature, and the true geometry is then anisotropic. The rejection condition re-introduces the anisotropy which was left out from the proposal density, by favoring (accepting with higher probability) those samples which fall along certain directions. Under Assumption B, the sectional curvatures of $M$ are bounded below by $-\kappa^2$. The rejection condition favors directions $s$ which meet curvatures closer to $-\kappa^2$ and tends to discard those directions with curvatures closer to positive values (incidentally, this justifies the name curvature-based rejection sampling).

Underlying this behavior is the following mathematical fact. If the curvatures along a certain direction $s$ are closer to $-\kappa^2$, then (along that direction) the volume comparison inequality (8) becomes sharper. In turn, the right-hand side of (14) becomes larger (closer to 1) and acceptance more probable.

This is due to a complementary volume comparison inequality, called Günther's inequality (Chavel, 2006) (Page 128), which implies that, if the sectional curvatures along the direction $s$ are bounded above by some negative $-\delta^2$,

$$|\det A(r, s)| \geq \left(\delta^{-1} \sinh(\delta r)\right)^{d-1} \tag{18}$$

It is precisely this lower bound which controls the sharpness of the upper bound in (8). The rejection condition favors directions which admit a $\delta$ closer to $\kappa$, because this makes (8) closer to being an equality, as it were.

In short, samples generated from the proposal density of CURS explore all directions with equal probability. However, directions which display "atypical cuvatures" are then gotten rid of by the rejection condition. On the other hand, it should be clear that the probability of rejection increases with the distance $r$, for samples which fall along the same direction. A concrete example of the above geometric interpretation will be provided in the following section (in particular, see the discussion after (27)).

## 5 Generalised Gaussian distributions

Recall the problem of sampling from the Gaussian density (2) on the space $M$ of real covariance matrices. Gaussian densities have found several applications since their introduction in (Said et al., 2018). Incidentally, they are the proposal densities required for the Riemannian proximal sampler from (Guan et al., 2025). Here, CURS is applied to the more general class of densities

$$p(x) \propto \exp\left[-\frac{d^\alpha(x_o, x)}{2\sigma^2}\right] \tag{19}$$

defined on the space $M$ of $N \times N$ real covariance matrices, for $\alpha > 1$ and $\sigma > 0$. In fact, two versions of CURS will be considered. First, the general version outlined in the Section 3, and then an improved version, more specific to the problem at hand.

Everything in the present section extends immediately, from the particular case where $M$ is the space of real covariance matrices to the general case where $M$ is any negatively-curved symmetric space (see Appendix D). This is because any $M$ which belongs to this general case satisfies the above Assumptions A, B, and C. In particular, this general case includes various spaces of covariance matrices (real, complex, Toeplitz, *etc.*), which are useful for an extensive range of applications (Said et al., 2018).

### 5.1 General CURS

The space $M$ is here equipped with its well-known affine-invariant Riemannian metric (Pennec et al., 2006),

$$\langle u, v \rangle_x = \text{tr}\left(x^{-1} u x^{-1} v\right) \tag{20}$$

for tangent vectors $u$ and $v$ to $M$ at $x$ (these are identified with real symmetric matrices, in the above formula). Therefore, one has the following Riemannian distance (Pennec et al., 2006)

$$d^2(x, y) = \text{tr}\left(\log\left(x^{-1/2} y x^{-1/2}\right)\right)^2 \tag{21}$$

where tr denotes the trace and log the symmetric matrix logarithm. In order to spell out the main steps of Algorithm 1, an additional bit of information is needed. Namely, this is (see Appendix B)

$$\det A(r,s) = r^{N-1} \prod_{i<j} \left( \sinh(\kappa_{ij}(s)r)/\kappa_{ij}(s) \right) \tag{22}$$

for the volume density in (6) — note this shows that Assumption C is here satisfied. Here,

$$\kappa_{ij}(s) = (\varsigma_i - \varsigma_j)/2 \quad 1 \leq i < j \leq N \tag{23}$$

where $(\varsigma_i)$ denote the eigenvalues of the symmetric matrix $s$. With this in mind, consider the following.

**step 1:** without any loss of generality, one may assume that $x_o = \mathrm{id}$, the $N \times N$ identity matrix. Then, the metric (20) becomes the usual trace metric (*i.e.* scalar product)

$$\langle u, v \rangle_{x_o} = \mathrm{tr}(uv) \tag{24}$$

An orthonormal basis for this metric is afforded by the matrices $u_{ij}$ $(1 \leq i \leq j \leq N)$,

$$\begin{aligned} u_{ij} &= (e_{ij} + e_{ji})/\sqrt{2} && \text{if } i < j \\ &= e_{jj} && \text{if } i = j \end{aligned}$$

where $e_{ij}$ is a matrix all of whose entries are zero, except the entry at row $i$ and column $j$, which is equal to 1. Accordingly, to carry out step 1, let $s_{ij}$ be independent standard normal real random variables and introduce

$$s = \sum_{i \leq j} s_{ij} u_{ij} \tag{25}$$

Then, replace $s$ with $s/\|s\|$, where $\|s\|^2 = \mathrm{tr}(s^2)$ since $\|s\|$ is the norm of $s$ with respect to (24).

Note here that it is possible to construct $s$ as in (25) without explicitly dealing with the matrices $(u_{ij})$. If $t$ is an $N \times N$ matrix with independent elements, each having standard normal distribution, it is enough to set $s = (t + t^\dagger)/2$, where $\dagger$ denotes transposition (in the language of random matrix theory, $s$ is then distributed according to the so-called Gaussian orthogonal ensemble (Livan et al., 2018)).

**step 2:** $r$ must be sampled from the density

$$g(r) \propto \exp(-r^\alpha/2\sigma^2) \left( \kappa^{-1} \sinh(\kappa r) \right)^{d-1} \tag{26}$$

where $\kappa = 1/\sqrt{2}$ and $d = N(N+1)/2$, the dimension of $M$. The fact that Assumption B is satisfied with $\kappa = 1/\sqrt{2}$ was mentioned in Section 2.2 and is proved in Appendix B.

Since $\alpha > 1$, the density (26) is a log-concave univariate density, and can be sampled using the universal black-box algorithm of (Devroye, 1986).

**step 4:** using (22) and also the fact that $\kappa = 1/\sqrt{2}$, it is straightforward to evaluate the condition (14). This becomes

$$U > \left( \kappa r/\sinh(\kappa r) \right)^{N-1} \prod_{i<j} \frac{(\sinh(\kappa_{ij}(s)r)/\kappa_{ij}(s))}{(\sinh(\kappa r)/\kappa)} \tag{27}$$

Recalling (23), it can be seen that this condition tends to reject those $s$ whose eigenvalues $(\varsigma_i)$ are too close to one another, and that rejection becomes more likely as $r$ increases. This is in agreement with the geometric interpretation of Section 4 (rejection of directions $s$ with curvatures closer to positive values), as can be seen from the principal curvatures formula (53), which is discussed in Appendix B.

Some care is required with the right-hand side of (27), as this involves the product of a potentially large number (equal to $d-1$) of small quantities. In practice, it is better to evaluate the condition after taking logarithms of both sides.

**step 8:** when $x_o = \mathrm{id}$, the Riemannian exponential mapping $\mathrm{Exp}_{x_o}$ in (3) is just the matrix exponential.

Finally, an additional step (step 9, then) is needed if $x_o \neq$ id. This re-centres samples $x$ around the new $x_o$ using the transformation $x \mapsto x_o^{1/2} \, x \, x_o^{1/2}$.

When $\alpha = 2$, the acceptance probability $\Pi$ in (15) can be computed analytically, thanks to the expressions from (Said et al., 2023; Santilli & Tierz, 2021). Here, $Z$ and $Z_\kappa$ are denoted $Z(\sigma)$ and $Z_\kappa(\sigma)$, in order to highlight their dependence on $\sigma$. Thus,

$$Z(\sigma) = (\pi\sigma^2/2)^{N/2} \, 2^{N(N-1)/4} \, \prod_{j=1}^{N} \Omega_{j-1} \times \exp\left[-N(N-1)^2(\sigma^2/8)\right] \times \mathrm{Pf}(\Lambda(\sigma)) \tag{28}$$

where $\Omega_{j-1}$ is the surface area of a $(j-1)$-dimensional sphere, and Pf denotes the Pfaffian, while $\Lambda(\sigma)$ is the $N \times N$ matrix whose elements are given by

$$\Lambda_{ij}(\sigma) = \exp\left[(i^2 + j^2)(\sigma^2/2)\right] \mathrm{erf}((j-i)(\sigma/2))$$

for $0 \leq i, j \leq N-1$, with erf the error function (the definition of the Pfaffian can be recalled from (Livan et al., 2018)). On the other hand,

$$Z_\kappa(\sigma) = \left(\frac{\sigma \, \Omega_{d-1}}{(2\kappa)^{d-1}}\right) \sum_{j=0}^{d-1} (-1)^j \, C_j^{d-1} \frac{\Phi((d-1-2j)\kappa\sigma)}{\Phi'((d-1-2j)\kappa\sigma)} \tag{29}$$

where $C_j^{d-1}$ is a binomial coefficient and $\Phi$ is the standard normal cumulative distribution function. It is important to note that (28) only holds when $N$ is even (there is a similar expression that can be used for odd $N$ (Santilli & Tierz, 2021)).

Thanks to Expressions (28) and (29), it is possible to match the empirical acceptance probability $\hat{\Pi}$, observed from multiple runs of CURS, with its theoretical counterpart $\Pi$, computed from (15). Agreement between theory ($\Pi$) and observation ($\hat{\Pi}$) is clear from the following Figures 1 and 2.

Figure 1 shows the plot of the theoretical acceptance probability $\Pi$ as a function of $\sigma$ (on the horizontal axis). This plot is overlaid with individual values of the empirical probability $\hat{\Pi}$, with each value estimated from $10^6$ iterations of the CURS algorithm (Algorithm 1). These are marked by dot points, which show no visible deviation away from the theoretical plot.

The acceptance probability clearly decreases with $\sigma$ and the rate of this decrease becomes more rapid as $N$ (the matrix dimension) increases. This is a manifestation of the curse of dimensionality. To put this effect into perspective, Figure 2 shows the dependence of $\Pi$ and $\hat{\Pi}$ on a new variable

$$\delta = \int_M d^2(x_o, x) \, p(x) \, \mathrm{vol}(dx) \tag{30}$$

the expected squared Riemannian distance away from $x_o$, with respect to the target density $p(x)$ in (19), with $\alpha = 2$. It is more intuitive to think in terms of $\delta$, since this new variable fits our usual idea of "variance".

For example, Figure 2b shows that when $N = 4$, for a variance $\delta = 2$, the acceptance probability is $\Pi \simeq 0.27$. Thus, on average, to obtain one new sample from the target density, 3.7 iterations of CURS are necessary. The actual time for a single iteration is of the order of $10^{-2}$ seconds (on a standard desktop computer).

By way of comparison, note that sampling from the same target distribution in (19) (as before, with $\alpha = 2$) was considered in (Bharath et al., 2025). Using a geometric Langevin algorithm, in the case where $N = 3$, this work reported having to compute upwards of $10^6$ trajectories of a discretised Langevin diffusion, in order to obtain a good approximation of the target density (see Table 3, in (Bharath et al., 2025)).

This requires a computational effort far greater than the one involved in CURS, since a single trajectory may contain hundreds of intermediate points, and computing each point requires evaluating both a matrix logarithm and a matrix exponential (see Algorithm 2, in (Bharath et al., 2025)). By contrast, one iteration of CURS evaluates only a matrix exponential, and this only once.

In all generality, one should also recall CURS is an exact sampling method, whereas the Langevin algorithm (being an MCMC method) is only approximate. To produce new samples, the Langevin algorithm either has to compute new trajectories, or to run a single trajectory for a sufficiently long time.

## 5.2 Sharp CURS

The acceptance probability observed in the previous subsection suffers from a significant drop as the matrix dimension $N$ increases. While this is to be expected, as rejection sampling methods tend to suffer from the curse of dimensionality, it can still be partially mitigated by employing a new, "sharper" version of CURS. This new version applies the same steps as in the previous subsection, but replaces (27) with the condition

$$U > \prod_{i<j} \frac{(\sinh(\kappa_{ij}(s)r)/\kappa_{ij}(s))}{(\sinh(\kappa r)/\kappa)} \tag{31}$$

When looked at closely, this is the same as (27), but without the factor $(\kappa r/\sinh(\kappa r))^{N-1}$ before the product. Since this factor quickly becomes small as $N$ increases, removing it should improve the acceptance probability.

Both conditions (27) and (31) aim to put upper bounds on the volume density (22). This density contains $d-1$ factors ($d = N(N+1)/2$ being the dimension of $M$). The first $N-1$ factors are all equal to $r$ and the remaining $N(N-1)/2$ factors are $\sinh(\kappa_{ij}(s)r)/\kappa_{ij}(s)$ where the $\kappa_{ij}(s)$ are given by (23) for $1 \le i < j \le N$.

Condition (27) is obtained by upper bounding each one of these $d-1$ factors by the same quantity $\sinh(\kappa r)/\kappa$. This is indeed a correct upper bound because $\kappa_{ij}(s) \le \kappa$, as explained in Appendix B after Formula (53). However, this upper bound is an exaggerated overestimate for the first $N-1$ factors, because $\sinh(\kappa r)/\kappa$ increases much faster than $r$.

Condition (31) upper bounds the first $N-1$ factors (all equal to $r$) by $r$, and maintains the upper bound $\sinh(\kappa r)/\kappa$ for the remaining $N(N-1)/2$ factors. A sharper inequality is thus obtained, whose effect is to remove $(\kappa r/\sinh(\kappa r))^{N-1}$ before the product from (27).

As before, it is helpful to have a meaningful geometric interpretation of the method at hand. In Section 4, it was explained that the idea of the CURS method is to reject directions which display "atypical curvatures".

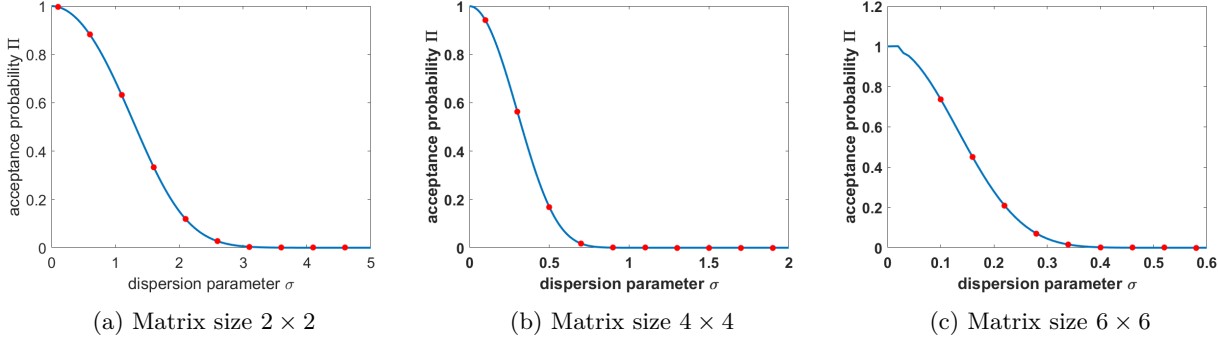

(a) Matrix size $2 \times 2$        (b) Matrix size $4 \times 4$        (c) Matrix size $6 \times 6$

Figure 1: Theoretical (solid line) and empirical (dot points) acceptance probabilities, plotted against $\sigma$

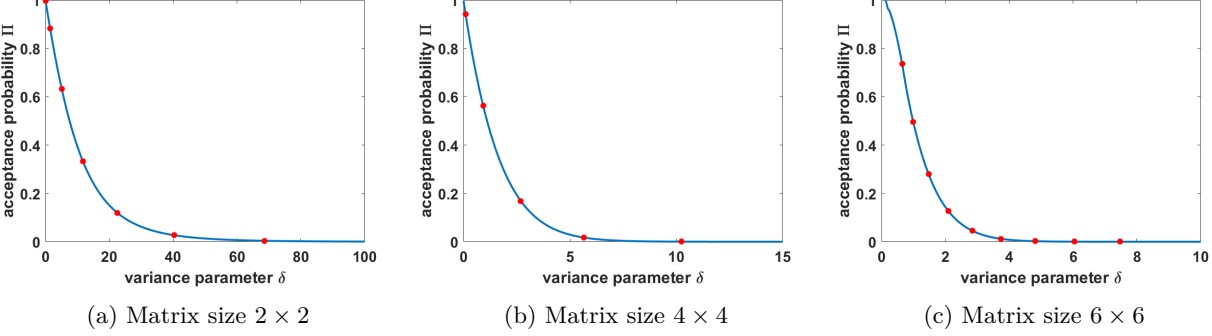

(a) Matrix size $2 \times 2$        (b) Matrix size $4 \times 4$        (c) Matrix size $6 \times 6$

Figure 2: Theoretical (solid line) and empirical (dot points) acceptance probabilities, plotted against $\delta$

Table 1: ($N = 4$) Empirical acceptance probabilities for CURS ($\hat{\Pi}$) and sharp CURS ($\hat{\Pi}_s$)

| $\sigma$ | 0.2 | 0.4 | 0.6 | 0.8 | 1.0 | 1.2 | 1.4 |
|---|---|---|---|---|---|---|---|
| $\hat{\Pi}$ | 0.7817 | 0.3430 | 0.0638 | 0.0031 | 0.0000 | 0 | 0 |
| $\hat{\Pi}_s$ | 0.8682 | 0.5510 | 0.2364 | 0.0606 | 0.0086 | 0.0006 | 0.0000 |

Table 2: ($N = 6$) Empirical acceptance probabilities for CURS ($\hat{\Pi}$) and sharp CURS ($\hat{\Pi}_s$)

| $\sigma$ | 0.1 | 0.2 | 0.3 | 0.4 | 0.5 | 0.6 | 0.7 |
|---|---|---|---|---|---|---|---|
| $\hat{\Pi}$ | 0.7377 | 0.2798 | 0.0449 | 0.0022 | 0.0000 | 0 | 0 |
| $\hat{\Pi}_s$ | 0.8067 | 0.4126 | 0.1224 | 0.0179 | 0.0011 | 0.0000 | 0 |

Now, the sectional curvatures encountered along a direction $s$ are determined by the principal curvatures (given by (53), Appendix B). Out of these $d - 1$ principal curvatures, there are always $N - 1$ equal to 0 (the others are negative). Since these $N - 1$ principal curvatures equal to 0 occur along every direction $s$, they can hardly be considered "atypical", even though they are different from the other principal curvatures which mostly have strictly negative values.

The main difference between general CURS (using Condition (27)) and sharp CURS (using Condition (31)), is that the former systematically discriminates against curvatures closer to positive values, whereas the latter (using specific information about the space of real covariance matrices) takes into account the existence of principal curvatures equal to 0.

Tables 1 and 2 show the improvement in acceptance probability obtained by using sharp CURS instead of CURS. These tables report empirical acceptance probabilities, each one estimated from $10^6$ iterations of either method. Note that standard deviations are not reported, since such a large number of iterations makes them negligible. When $N = 4$ and $\sigma = 1$, Table 1 shows that the acceptance probability for CURS is less than $10^{-4}$, while sharp CURS has acceptance probability $86 * 10^{-4}$ — a rather large improvement. Thus, sharp CURS produces a new sample every $\approx 116$ iterations on average. Each iteration requires a time of the order of $10^{-2}$ seconds. For $N = 4$ and $\sigma = 1$ the variance $\delta$ in (30) is a pretty big 13.3 (compare to discussion in the previous subsection). This can be found from $\delta = \sigma^3 Z'(\sigma)/Z(\sigma)$ with $Z(\sigma)$ given by (28) (Said et al., 2023) (the prime denotes the derivative). Table 2 reports a very similar situation when $N = 6$. For example, at $\sigma = 0.5$, the variance is $\delta = 7.47$. The acceptance probability for CURS is less than $10^{-4}$, while it is $11 * 10^{-4}$ for sharp CURS. The time required for a single iteration (of either method) is still of the order of $10^{-2}$ seconds.

Experiments carried out in the present section have completely focused on the Gaussian case ($\alpha = 2$) in (19). The situation with other values of $\alpha$ is briefly discussed in Appendix E.

### 5.3   A look at the samples

By construction, CURS (or sharp CURS) is an exact sampling method. It produces independent samples which exactly follow the target distribution. Still, it is helpful to have a look at generated samples and verify they behave as expected. Here, this is done for samples obtained with sharp CURS from a Gaussian distribution (19) with $\alpha = 2$, as in the previous paragraph. First, a visual verification is considered in the case of $2 \times 2$ matrices ($N = 2$). When a $2 \times 2$ real covariance matrix $x$ is distributed according to (19) with $\alpha = 2$, its log-eigenvalues have a well-known joint probability density Said et al. (2018). Specifically, if the eigenvalues of $x$ are denoted $\exp(r_1)$ and $\exp(r_2)$ with $r_1 \leq r_2$, then the joint probability density of $(r_1, r_2)$ reads

$$p(r_1, r_2) \propto \exp\left(-(r_1^2 + r_2^2)/2\sigma^2\right) \sinh\left((r_2 - r_1)/2\right) \mathbf{1}_{\{r_1 \leq r_2\}} \tag{32}$$

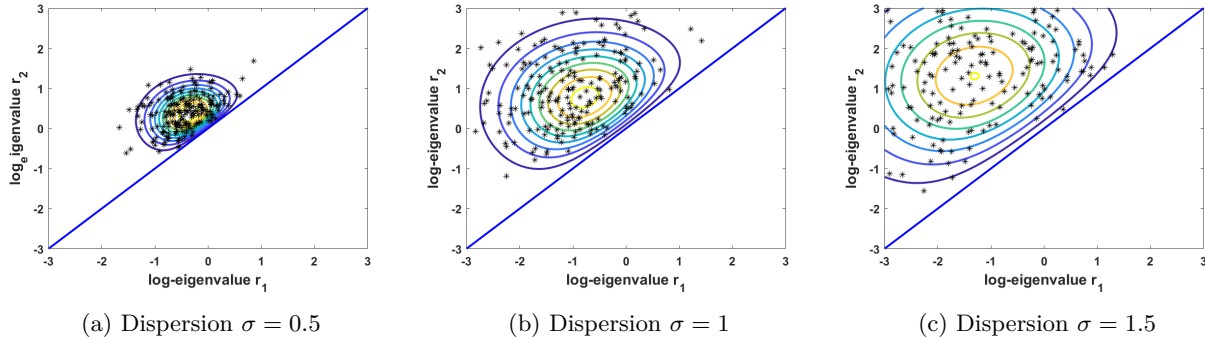

(a) Dispersion $\sigma = 0.5$         (b) Dispersion $\sigma = 1$         (c) Dispersion $\sigma = 1.5$

Figure 3: Sharp CURS at $N = 2$: scatter plot of log-eigenvalues under Gaussian distribution (19) with $\alpha = 2$

where $\mathbf{1}_{\{r_1 \leq r_2\}}$ is the indicator function of the set $\{r_1 \leq r_2\}$.

Figure 3 exhibits contour plots of this joint density, overlaid with a scatter plot of log-eigenvalues obtained by diagonalising 200 samples (matrices) generated using sharp CURS. These are seen to hug the density contours, just as one may expect. Note that the joint density (32) vanishes along the main diagonal $r_1 = r_2$ (the main diagonal is not part of the contour plot, but was added to the figures for emphasis).

Consider a higher-dimensional experiment, with $N = 4$. The aim is to recover the parameters $x_o$ and $\sigma$, by applying maximum-likelihood estimation to samples generated from $10^6$ iterations of the sharp CURS algorithm. This was done with $x_o = \mathrm{id}$ ($4 \times 4$ identity matrix) and with $\sigma$ ranging between 0 and 1.5. As $\sigma$ increases, the acceptance probability drops, and fewer samples are available. The worst case situation (at $\sigma = 1.5$) still had an acceptance probability of about $1.3 * 10^{-4}$, which leaves 130 samples to work with.

The maximum-likelihood estimate of $x_o$ is the Riemannian centre of mass of the samples, which can be computed through a standard Riemannian gradient descent procedure Said et al. (2018). The distance $d(\hat{x}_o, x_o)$ between the maximum-likelihood estimate $\hat{x}_o$ and the true $x_o = \mathrm{id}$ remains of the order of $10^{-4}$ as long as $\sigma \in (0, 1]$. At $\sigma = 1.5$ it rises to $0.0743 \pm 0.0330$ (mean and standard deviation taken over ten trials). This shows the samples produced by sharp CURS allow the parameter $x_o$ to be recovered correctly.

The maximum-likelihood estimate of $\sigma$ is found by computing an empirical estimate $\hat{\delta}$ of the variance $\delta$ in (30), and then matching with the theoretical $\delta = \sigma^3 Z'(\sigma)/Z(\sigma)$ with $Z(\sigma)$ given by (28) (Said et al., 2023) (the prime denotes the derivative).

The following Table 3 indicates the theoretical value of $\delta$ (denoted $\delta(\sigma)$) and the corresponding empirical $\hat{\delta}$. These two match very closely, except for larger values of $\sigma$ where the acceptance probability is too weak. It is clear from this table the samples procuded by sharp CURS allow for $\sigma$ to be recovered accurately.

The above visual verification at $N = 2$ and maximum-likelihood estimation experiment at $N = 4$, while providing solid evidence that samples generated by sharp CURS do exactly follow the target distribution, do not constitute rigorous goodness of fit tests. A kernel-based goodness of fit test, for densities on the manifold of real covariance matrices, can (for example) be developed by analogy with the two-sample hypothesis test in Said et al. (2025). However, this falls outside the scope of the present work.

Table 3: Sharp CURS at $N = 4$: $\delta(\sigma)$ compared to $\hat{\delta}$ under Gaussian distribution (19) with $\alpha = 2$

| $\sigma$ | 0.2 | 0.4 | 0.6 | 0.8 | 1.0 | 1.2 | 1.4 |
|---|---|---|---|---|---|---|---|
| $\delta(\sigma)$ | 0.4048 | 1.6782 | 4.0047 | 7.7163 | 13.3238 | 21.5492 | 33.3494 |
| $\hat{\delta}$ | 0.4050 | 1.6797 | 4.0053 | 7.6978 | 13.3047 | 22.4178 | 34.2023 |

## 6  Dealing with the cut locus

The present section discusses dropping the simplifying Assumption A, made in Subsection 2.1. This was the assumption that geodesic spherical coordinates regularly cover the entire manifold $M$, so that the domain of injectivity $D_{x_o}$ satisfies $D_{x_o} = M$.

In general, the manifold $M$ is the disjoint union of $D_{x_o}$, which is an open subset of $M$, and of the so-called cut locus $\text{Cut}_{x_o}$. This cut locus is a closed subset of $M$ and has zero Riemannian volume (Chavel, 2006). Now, the target distribution (4) clearly assigns zero probability to any subset of $M$ that has zero volume. In particular, the probability that a sample $x$ should belong to $\text{Cut}_{x_o}$ is (according to (4))

$$P(\text{Cut}_{x_o}) = \int_{\text{Cut}_{x_o}} P(dx) = \int_{\text{Cut}_{x_o}} p(x)\,\text{vol}(dx) = 0$$

where the last equality follows from the fact that $\text{Cut}_{x_o}$ has zero volume. For the sampling problem at hand, this has the following implication: it is possible to consider the target distribution (4) (with its density (1)) as a distribution on the injectivity domain $D_{x_o}$ and thereby to restrict sampling to this open subset of $M$.

This allows for the CURS algorithm to be extended to the fully general situation, where one has $D_{x_o} \neq M$, but at the cost of an additional modification. In fact, the final step of Algorithm 1 (step 8) produces a sample $x$ by replacing $(r, s)$ into the exponential map in (3). As of now, the couple $(r, s)$ should be restricted in such a way that (3) will only produce samples $x$ that belong to $D_{x_o}$. This restriction is indeed necessary. If it is not applied, the exponential map is not anymore a diffeomorphism (it may even fail to be bijective).

To introduce the correct restriction, one needs to explicitly know the so-called $c$-function (Chavel, 2006), $c : S_{x_o}M \to (0, \infty)$, where $c(s)$ is the least value of $r > 0$ for which $x(r, s)$ (given by (3)) does not belong to $D_{x_o}$. Using this function, the target joint density $p(r, s)$ from (9) is now replaced with the truncated version

$$p_c(r, s) = \frac{1}{Z} \times f(r)\,|\det A(r, s)| \times \mathbf{1}_{\{r < c(s)\}} \tag{33}$$

where $Z$ is a normalising constant, $\mathbf{1}_{\{r < c(s)\}}$ denotes the indicator function of the set $\{r < c(s)\}$, and the truncated density $p_c$ is with respect to the reference measure $dr\,\omega(ds)$.

In this new setting, the task of CURS is to sample $(r, s)$ from the target joint density (33) and then to obtain $x$ by replacing into (3). The general idea introduced in Subsection 2.2 can be employed once more. Namely, using a volume comparison inequality, obtain an upper bound on the target joint density (33), and then use this to perform rejection sampling. However, it should be emphasised that this idea will only be applicable under the following assumption.
**Assumption D** the $c$-function is known explicitly and $c(s)$ can be readily evaluated for any $s \in S_{x_o}M$.
As a concrete example of a Riemannian manifold $M$ which satisfies this assumption, consider $M = U(N)$, the group of $N \times N$ unitary matrices, equipped with the trace metric

$$\langle u, v \rangle_x = \text{tr}\left(uv^\dagger\right) \tag{34}$$

where $\dagger$ denotes conjugation-transposition and the tangent vectors $u$ and $v$ at $x \in M$ are $N \times N$ matrices, such that $x^\dagger u$ and $x^\dagger v$ are skew-Hermitian.

Here is a detailed discussion of the steps involved in applying CURS for this particular manifold.

Consider a target density of the general form (1), and assume without loss of generality that $x_o = \text{id}$, the $N \times N$ identity matrix. Then, to express the truncated density $p_c$ from (33), note the following facts (see Appendix C for background).

● the tangent space to $M$ at $x_o = \text{id}$ is the space of $N \times N$ skew-Hermitian matrices. The unit sphere $S_{x_o}M$ is then made up of those skew-Hermitian $s$ with $\|s\|^2 = 1$, where the norm is determined by the metric (34).

● for $r > 0$ and $s \in S_{x_o}M$, the volume density is given by

$$\det A(r, s) = r^{N-1} \prod_{i < j} \left(\sin(\kappa_{ij}(s)r)/\kappa_{ij}(s)\right)^2 \tag{35}$$

where, similar to (23),

$$\kappa_{ij}(s) = (\varsigma_i - \varsigma_j)/2 \qquad 1 \leq i < j \leq N \tag{36}$$

with the eigenvalues of $s$ being $j\varsigma_i$ for $i = 1, \ldots, N$ (here, $j = \sqrt{-1}$).

• the $c$-function is given in the following manner, for each $s \in S_{x_o}M$,

$$c(s) = \frac{\pi}{\max_i |\varsigma_i|} \tag{37}$$

Formulas (35)–(37) completely determine the truncated density $p_c$ in (33), with the normalising constant

$$Z = \int_{S_{x_o}M} \int_0^{c(s)} f(r) \, |\det A(r,s)| \, dr\omega(ds) \tag{38}$$

As stated above, the job of CURS is to sample from this truncated density. In order to apply the idea of Subsection 2.2, one has to upper bound $p_c(r, s)$ using some other density $g(r, s)$ which does not depend on $s$. The chosen upper bound should reflect knowledge of the curvature of the manifold $M$ at hand. Specifically, this is a manifold with positive sectional curvatures (Chavel, 2006), so the volume comparison inequality (8) should be applied with $\kappa = 0$, which yields $|\det A(r,s)| \leq r^{d-1}$ ($d = N^2$ is the dimension of $M$).

Replacing this last inequality into (33), and noting that the indicator function of $\{r < c(s)\}$ is always $\leq 1$, one is immediately lead to

$$p_c(r, s) \leq T \times g(r, s) \tag{39}$$

in terms of the following

$$g(r, s) = \frac{1}{Z_0} \times f(r)r^{d-1} \times \mathbf{1}_{\{r < \Delta_{x_o}\}} \qquad T = \frac{Z_0}{Z} \tag{40}$$

where $Z_0$ is a normalising constant, and $\Delta_{x_o} = \max c(s)$ (where the maximum is taken over $s \in S_{x_o}M$). Note that $\Delta_{x_o}$ is the greatest possible Riemannian distance $d(x_o, x)$, between $x_o$ and any other point $x \in M$. Having obtained (39) and (40), it is possible to spell out the steps of a CURS algorithm for sampling on the manifold $M = U(N)$. While specific to this manifold, these steps hopefully offer a clear first idea of what a fully general CURS algorithm should look like.

**step 1 :** as in Subsection 5.1, the first step is to sample $s$ from a uniform distribution on the sphere $S_{x_o}M$. To do so, let $s = jh$ where $h$ is distributed according to the Gaussian unitary ensemble, and then replace $s$ by $s/\|s\|$ (where the norm is determined by (34)). Note that $h = (t + t^\dagger)/2$ where $t$ has independent entries, $t_{ij} = a_{ij} + jb_{ij}$ with $a_{ij}$ and $b_{ij}$ independent, Gaussian, each with mean 0 and variance $1/2$ (Forrester, 2010).

**step 2 :** again as in Subsection 5.1, sample the distance $r$ from the proposal density $g(r) = g(r, s)$ in (40). However (unlike Subsection 5.1), then reject any couple $(r, s)$ such that $r \geq c(s)$, where $c(s)$ is given by (37). For (40), note that $\Delta_{x_o} = \sqrt{N}\pi$ (see Appendix C).

**step 3 :** for the rejection procedure, sample $U$ from a uniform distribution in the unit interval $(0, 1)$.

**steps 4 and 5 :** reject $(r, s)$ if

$$U > \prod_{i<j} \left( \sin(\kappa_{ij}(s)r)/\kappa_{ij}(s)r \right)^2 \tag{41}$$

Once these four steps lead to acceptance of some couple $(r, s)$, it is enough to put $x = \exp(rs)$ where $\exp$ denotes the matrix exponential. This amounts to applying the general formula (3). Moreover, if $x_o \neq$ id, it is also necessary to re-centre the sample $x$ around $x_o$ using the transformation $x \mapsto x_o x$.

The above described steps closely mirror those in Subsection 5.1. The only difference appears in step 2, which has an additional rejection step that discards $(r, s)$ when $r \geq c(s)$. The manifold $M = U(N)$ is compact, and the Riemannian distance between any two points $x_o$ and $x$ cannot exceed $\sqrt{N}\pi$ (the diameter of $M$). If only for this reason, one cannot accept to sample a distance $r$ that is too large. Furthermore, without the restriction $r < c(s)$, the exponential map will fail to be a diffeomorphism, so that taking $x = \exp(rs)$ will not produce correct samples from the target distribution.

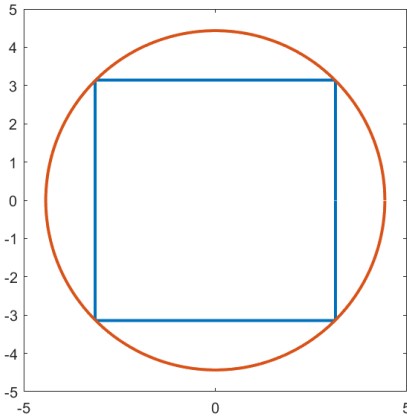

Figure 4: Proposal and acceptance regions for CURS ($M = U(2)$)

Steps 3 and 4 look just like the corresponding steps in Section 5.1 but the rejection condition (41) works in quite a different way from (27) (and also from (31)). Condition (41) favors directions $s$ which have eigenvalues ($j\varsigma_i$) close to one another (these have small $\kappa_{ij}(s)$, by (36)), while the previous (27) tends to reject $s$ with close eigenvalues. This is a striking difference between how CURS works for spaces of positive curvature (the present case) and spaces of negative curvature (Section 5).

A complete analysis of the above CURS algorithm for sampling on $M = U(N)$ would require evaluation of the acceptance probability $\Pi = 1/T$ (where $T$ is given by (40)). This will not be carried out here, since the mathematical tools required are somewhat outside of the scope of the present work.

It is helpful to have a feeling for (or intuition about) the acceptance region $r < c(s)$. When $N = 2$, this can be obtained through visualisation in a two-dimensional plane, with the coordinates $x_1 = r\varsigma_1$ and $x_2 = r\varsigma_2$, as in Figure 4. In this figure, the interior square encloses the acceptance region $r < c(s)$, while the outer circle corresponds to $r = \Delta_{x_o}$ (this is equal to $\sqrt{2}\pi$, since $N = 2$). In step 2 (which was just described), the proposal density produces samples inside the circle, but only those that fall inside the square are retained.

The samples which fall inside the square are then submitted to the rejection condition (41), whose right-hand side is just $(\text{sinc}((x_1 - x_2)/2))^2$ (where $\text{sinc}(x) = \sin(x)/x$). This is equal to 1 when $x_1 = x_2$, so samples which fall along the main diagonal have a probability equal to 1 of being accepted. The probability of acceptance decreases along the antidiagonal, and becomes zero at the corners of the square, $(x_1, x_2) = (-\pi, \pi)$ or $(\pi, -\pi)$.

## 7 Summary and conclusion

Curvature-based rejection sampling (CURS) is a new method for sampling from a probability density $p$ defined on a Riemannian manifold $M$. It is a broadly applicable method, since it only requires two essential conditions. First, the density $p$ "depends only on distance", which means that there exists some point $x_o \in M$ such that

$$p(x) \propto f(r(x)) \text{ where } r(x) = d(x_o, x) \tag{42}$$

where $f$ is some function and $d(x_o, x)$ denotes the Riemannian distance between $x_o$ and $x$ in $M$. Second, the sectional curvatures of the Riemannian manifold $M$ are always $\geq -\kappa^2$ for some $\kappa \geq 0$.

The idea behind CURS is to combine the geometric principle of volume comparison with the statistical principle of rejection sampling. The density $p(x)$ can be expressed in geodesic spherical coordinates $(r, s)$ ($r$ is just the above $r(x)$ and the unit vector $s$ points from $x_o$ to $x$), where it reduces to the joint density

$$p(r, s) = f(r) \left| \det A(r, s) \right| \tag{43}$$

where $\left| \det A(r, s) \right|$ is the volume density. Bishop's volume comparison theorem provides the upper bound

$$\left| \det A(r, s) \right| \leq \left( \kappa^{-1} \sinh(\kappa r) \right)^{d-1} \qquad d = \dim M \tag{44}$$

Remarkably, this inequality is completely independent of the specific Riemannian manifold $M$. Therefore, replacing it into (43) allows one to implement a straightforward rejection sampling strategy on any manifold $M$ with sectional curvatures $\geq -\kappa^2$.

CURS is an exact sampling method, and has a quite moderate computational cost, especially in comparison with methods based on MCMC strategies (*e.g.* Hamiltonian Monte Carlo or geometric Langevin MCMC). Its major drawback is the fact that its performance is significantly affected by the curse of dimensionality. The aim of the present work has been to introduce this new method and to advocate its use in low-dimensional scenarios, where it offers a pretty attractive performance-complexity tradeoff.

In addition, the CURS algorithm (Algorithm 1 of Section 3) is very easy to implement and has a clear geometric interpretation.

In the present work, CURS was applied to the problem of sampling from a generalised Gaussian distribution, defined on a space of real covariance matrices. For this problem, detailed numerical experiments were carried out. First, these experiments showed the performance of CURS (as measured by its acceptance probability) matches exactly with theoretical predictions. Second, they clearly illustated the noteworthy improvement in computational complexity which CURS achieves, in comparison with a geometric Lanvegin algorithm which has been applied to the same problem (by (Bharath et al., 2025)).

Further discussion, of how the CURS method can be concretely implemented, was provided for the case where the underlying manifold $M$ is a unitary group (of any dimension). This particular example was chosen to demonstrate how CURS deals with one of the main issues of sampling on Riemannian manifolds : the existence of a non-empty cut locus. Roughly, the cut locus is the subset of a Riemannian manifold where classical Riemannian coordinate systems (such as geodesic spherical coordinates) become singular or ill-defined. Things become quite complicated, as one cannot directly resort to sampling in terms of coordinates. It was shown the CURS methodology naturally incorporates a means of dealing with this difficult issue.

Hopefully, CURS will find its intended place among the many existing methods for sampling on manifolds. This method is based on an elegant theoretical principle, is easy to implement, and was shown to have a moderate complexity, coupled with a strong performance, as long as the curse of dimensionality does not get in its way.

Perhaps the key to the simplicity and generality of CURS is that, at its heart, lies what is arguably the fundamental concept of Riemannian geometry : curvature. In other words, this method does not immediately jump into the specific geometry of the manifold at hand, but first attempts to deal with it through general knowledge of its curvature properties.

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

## A   Integration in spherical coordinates

The aim here is to explain how the integral

$$\int_M f(x)\,\mathrm{vol}(dx) \tag{45}$$

where $f : M \to \mathbb{R}$ and vol denotes the Riemannian volume, can be expressed in geodesic spherical coordinates, introduced in Subsection 2.1. The following material is based on (Chavel, 2006) (Chapter III).

Let $M$ be a complete Riemannian manifold and $x_o \in M$. Then, $M$ is the disjoint union of the open subset $D_{x_o}$, called the domain of injectivity, and of the closed subset $\mathrm{Cut}_{x_o}$, called the cut locus.

Now, the cut locus has zero Riemannian volume, so the integral (45) may be restricted to its complement

$$\int_M f(x)\,\mathrm{vol}(dx) = \int_{D_{x_o}} f(x)\,\mathrm{vol}(dx) \tag{46}$$

In addition, the mapping $x \mapsto x(r,s)$ from (3) becomes a diffeomorphism onto $D_{x_o}$ when the couple $(r,s)$ is restricted in a certain way. This restriction is given by the so-called $c$-function (Chavel, 2006) (Page 115). This is $c : S_{x_o}M \to (0, \infty)$, where $c(s)$ is the least value of $r > 0$ for which $x(r,s)$ does not belong to $D_{x_o}$.

Then, integration over $D_{x_o}$ can be replaced by integration over $(r,s)$ such that $r < c(s)$. From (46),

$$\int_M f(x)\,\mathrm{vol}(dx) = \int_{D_{x_o}} f(x)\,\mathrm{vol}(dx)$$
$$= \int_{S_{x_o}M} \int_0^{c(s)} f(r,s)\,\mathrm{vol}(dr \times ds) \tag{47}$$

where $f(r,s) = f(x(r,s))$ and $\mathrm{vol}(dr \times ds)$ is just $\mathrm{vol}(dx)$, but expressed in terms of $r$ and $s$.

This was given by (6) in Subsection 2.1, which reads

$$\mathrm{vol}(dr \times ds) = |\det A(r,s)|\, dr\omega(ds)$$

where $\omega$ is the surface area measure on the unit sphere $S_{x_o}M$ and the matrix $A(r,s)$ is obtained by solving a second-order linear differential equation, known as the matrix Jacobi equation (Chavel, 2006) (Page 114).

Replacing into (47) gives the general integral formula in geodesic spherical coordinates

$$\int_M f(x)\,\mathrm{vol}(dx) = \int_{S_{x_o}M} \int_0^{c(s)} f(r,s)\,|\det A(r,s)|\, dr\omega(ds) \tag{48}$$

Formulas (16) and (17) in Section 3 are both special cases of (48). Indeed, (16) is just the same as (48), but with $c(s) = \infty$. This reflects the assumption, made in Section 2, that $D_{x_o} = M$. On the other hand, (17) is a consequence of (16), which follows by putting

$$\det A(r,s) = \left(\kappa^{-1}\sinh(\kappa r)\right)^{d-1} \tag{49}$$

which does not depend on $s$, and then integrating over $s$, which leads to the factor $\Omega_{d-1}$. In (16) and (17), by comparison with (48), $f(r,s) = f(r)$ does not depend on $s$.

The key to using (48) is the matrix Jacobi equation. Note that $A(r,s)$ is a $(d-1)\times(d-1)$ matrix ($d = \dim M$), and the matrix Jacobi equation reads (the prime denotes differentiation with respect to $r$)

$$A''(r,s) + R(r,s)\,A(r,s) = 0 \qquad A(0,s) = 0 \text{ and } A'(0,s) = \mathrm{I}_{d-1} \tag{50}$$

where $R(r,s)$ is a matrix given in terms of the Riemann curvature tensor of $M$ (Chavel, 2006) (Page 114), and $\mathrm{I}_{d-1}$ is the $(d-1) \times (d-1)$ identity matrix. For example, if $M$ has constant negative curvature $-\kappa^2$, then $R(r,s) = -\kappa^2 \mathrm{I}_{d-1}$, and the solution of (50) is $A(r,s) = \kappa^{-1}\sinh(\kappa r)\mathrm{I}_{d-1}$. This immediately gives (49).

# B Background for Section 5

The present appendix provides some additional background on the Riemannian geometry of the space $M$, of $N \times N$ real covariance matrices, equipped with the affine-invariant metric (20). Essentially, the aim is to apply the integral formula (48) to this space, and this will require solving the matrix Jacobi equation (50).

The main ingredient in this equation is the Riemann curvature tensor. Recall that the tangent space to $M$ at $x_o = \mathrm{id}$ is identified with the vector space of $N \times N$ real symmetric matrices. Denote this space by $T_{x_o}M$. The Riemann curvature tensor is a map $R : T_{x_o}M \times T_{x_o}M \times T_{x_o}M \to T_{x_o}M$, which is linear in each one of its three arguments (Lee, 2018). In the present case, it is given by (Helgason, 2001),

$$R(u,v)w = [w, [u,v]] \tag{51}$$

where $[a,b] = ab - ba$ for any matrices $a, b$.

The space $M$ at hand is moreover a symmetric space. In particular, its curvature tensor is parallel (Helgason, 2001), and this implies the matrix $R(r,s)$ in (50) is constant (does not depend on $r$). Moreover, it is just the matrix (in any chosen basis of $T_{x_o}M$) of the radial curvature operator (Lee, 2018),

$$R_s(u) = R(s,u)s \tag{52}$$

This operator $R_s : T_{x_o}M \to T_{x_o}M$ is diagonalisable, with the following eigenvalues: 0 with multiplicity $N$, and $-(\kappa_{ij}(s))^2$ (given by (23)) for $1 \le i < j \le N$, with multiplicity 1 (see (Said, 2021) for details). In addition, the corresponding eigenvectors form an orthonormal basis of $T_{x_o}M$.

Let $(u_\alpha; \alpha = 1, \ldots, d)$ be such an orthonormal basis, where each $u_\alpha$ corresponds to the eigenvalue $\lambda_\alpha$ ($d = N(N+1)/2$ is the dimension of $M$). In this basis, the matrix $A(r,s)$ (the solution of (50)) is diagonal, and its diagonal elements are solutions to ordinary differential equations,

$$A_{\alpha\alpha}''(r,s) = -\lambda_\alpha \, A_{\alpha\alpha}(r,s) \qquad A_{\alpha\alpha}(0,s) = 0 \text{ and } A_{\alpha\alpha}'(0,s) = 1$$

For the $N$ eigenvalues $\lambda_\alpha = 0$, this yields $A_{\alpha\alpha}(r,s) = r$. For the remaining eigenvalues $\lambda_\alpha = -(\kappa_{ij}(s))^2$, one has $A_{\alpha\alpha}(r,s) = \sinh(\kappa_{ij}(s)r)/\kappa_{ij}(s)$. Multiplying these together yields

$$\det A(r,s) = r^N \prod_{i<j} \left( \sinh(\kappa_{ij}(s)r)/\kappa_{ij}(s) \right)$$

The volume density (22) is obtained by replacing $r^N$ with $r^{N-1}$, since one of the eigenvectors with $\lambda_\alpha = 0$ is $u_\alpha = s$, which is normal to the sphere $S_{x_o}M$ and should not be included when integrating over this sphere.

Assume that $u_1 = s$, so each $u_\alpha$ with $\alpha > 1$ is orthogonal to $s$. Then, the vectors $s$ and $u_\alpha$ span a two-dimensional plane with sectional curvature (Said, 2021)

$$\sec(s, u_\alpha) = \lambda_\alpha = 0 \text{ or } -(\kappa_{ij}(s))^2 \tag{53}$$

These are known as principal sectional curvatures, and the sectional curvatures of $M$ at any point are included between the minimum and maximum of these $\lambda_\alpha$. It is clear from (23) that those $s$ which have eigenvalues ($\varsigma_i$) closer to one another encounter sectional curvatures closer to 0.

The sectional curvatures of $M$ are always negative or zero, by (53). The least possible sectional curvature is

$$-\kappa^2 = \min \, -(\kappa_{ij}(s))^2$$

where the minimum is over $s \in S_{x_o}M$. A straightforward differentiation shows that this minimum is achieved when there are two eigenvalues $\varsigma_i = -\varsigma_j$ both with absolute value $|\varsigma_i| = |\varsigma_j| = 1/\sqrt{2}$. This implies that $-\kappa^2 = -(\kappa_{ij}(s))^2 = -1/2$, and thus justifies the value of $\kappa = 1/\sqrt{2}$ used in Section 5.

Note that the presentation in this appendix is completely specific to the space of real covariance matrices. The following appendix puts it in a more general context, by considering the application of CURS to Riemannian symmetric space of non-compact type.

## C  Background for Section 6

The present appendix provides a background discussion for the facts used in Section 6 in order to express the truncated density (33) when $M = U(N)$.

The first one of these three facts states that the tangent space to $M = U(N)$ at $x_o = \mathrm{id}$ is the space of $N \times N$ skew-Hermitian matrices. This is a well-known statement in the elementary theory of Lie groups, and a short proof may be found (for example) in (Hall, 2015) (Chapter 3).

The second fact is the expression (35) for the volume density. This can be derived following a reasoning analogous to the one sketched in Appendix B, for the space of real covariance matrices. Indeed, $M = U(N)$ is also a Riemannian symmetric space (because it is a compact Lie group (Helgason, 2001)). Its curvature tensor is therefore parallel, and the matrix $R(r, s)$ in the matrix Jacobi equation (50) is constant.

The eigenvalues of this matrix, which are just the same as those of the radial curvature operator (52), are the following: 0 with multiplicity $N$, and $(\kappa_{ij}(s))^2$ (given by (36)) for $1 \leq i < j \leq N$, each with multiplicity 2 (details can be found in (Said, 2021)).

Knowledge of these eigenvalues leads to the solution of the matrix Jacobi equation (50), as in Appendix B. For the determinant $\det A(r, s)$, each zero eigenvalue contributes a factor $r$, while each eigenvalue $(\kappa_{ij}(s))^2$ contributes a factor $\sin(\kappa_{ij}(s)r)/\kappa_{ij}(s)$ — note that since these eigenvalues are positive, rather than negative as in Appendix B, there is a sine function instead of the hyperbolic sine function.

Upon multiplying these factors together,

$$\det A(r, s) = r^N \prod_{i<j} \left( \sin(\kappa_{ij}(s)r)/\kappa_{ij}(s) \right)^2$$

Then, (35) is recovered by replacing $r^N$ with $r^{N-1}$. This is for the same reason as in Appendix B. Namely, one of the eigenvectors corresponding to an eigenvalue equal to zero happens to be $s$, which should not counted in the volume density.

Finally, the third fact is the expression (37) for the $c$-function. This can be found in (Sakai, 1977), which computes $c(s)$ using the characterisation that $c(s)$ is the least value of $r > 0$ for which there exists $s' \in S_{x_o}M$, different from $s$, such that $x(r, s) = x(r, s')$ (in the notation of (3)). In other words, $c(s)$ marks the first time the geodesic curve starting at $x_o$ with initial velocity $s$ intersects another geodesic curve also starting at $x_o$.

The sectional curvatures of $M = U(N)$ belong to the interval between the minimum and maximum of the eigenvalues 0 and $(\kappa_{ij}(s))^2$ of the radial curvature operator. Clearly, then, this manifold $M$ has non-negative curvature, and the maximum sectional curvature is $\max (\kappa_{ij}(s))^2$ (with the maximum taken over $s \in S_{x_o}M$). It is possible to show this maximum is equal to $1/2$.

The diameter of $M = U(N)$ is the greatest possible Riemannian distance between two points $x_o$ and $x$ in $M$. This is $\Delta_{x_o} = \max c(s)$ with $c(s)$ given by (37). This maximum occurs when all $\varsigma_i$ are equal, and therefore equal to $1/\sqrt{N}$. This yields the diameter $\sqrt{N}\pi$. In fact, if $x_o = \mathrm{id}$ then the maximum Riemannian distance $d(x_o, x)$ occurs for $x = -\mathrm{id}$, and only for this point.

## D  CURS for symmetric spaces

The development in Section 5 and Appendix B directly generalises to a large class of Riemannian manifolds. This is the class of negatively-curved Riemannian symmetric spaces, of which the space of real covariance matrices is just one instance, but which also includes spaces of complex, quaternion, Toeplitz or block Toeplitz covariance matrices (Said et al., 2018).

The present appendix presents a general version of CURS, for negatively-curved Riemannian symmetric spaces. For now, the general treatment for positively-curved spaces (generalising Section 6 and Appendix C) will not be detailed. This is just to avoid making the present paper too long, and to focus on general ideas rather than technical details.

Now, let $M$ be a negatively-curved Riemannian symmetric space (Helgason, 2001). In particular, $M$ is a Hadamard manifold and therefore satisfies the simplifying assumption of Subsection 2.1 (that $D_{x_o} = M$). Therefore, application of CURS on this space reduces to the steps described in Algorithm 1 of Section 3.

Application of these steps requires knowledge of three things: (a) the lower bound $-\kappa^2$ on the sectional curvatures of $M$, (b) the volume density $\det A(r, s)$ (*e.g.* as in (22)), (c) finally (although this is not part of the algorithm in itself) the normalising constants $Z$ and $Z_\kappa$ which are needed to evaluate the acceptance probability $\Pi$ in (15).

To describe these three things, the most important mathematical objects will be the so-called positive restricted roots of the Lie algebra of the group of isometries of $M$ (Helgason, 2001). These will first be presented in general form, and then discussed through basic examples.

To begin, recall that $M$ admits a transitive action of a group of isometries $G$. Each isometry $g \in G$ is a mapping $g : M \to M$ that preserves the Riemannian metric of $M$. The subgroup of $G$, made up of those $g \in G$ that leave invariant the point $x_o \in M$, in the sense that $g(x_o) = x_o$ (the notation is that of (1)), will be denoted $K$.

Without any loss of generality, it is possible to think of the group $G$ as a matrix Lie group (Hall, 2015). This means that $G$ is a closed subgroup of the group $GL(q, \mathbb{R})$ of invertible $q \times q$ matrices (for some $q \geq 2$). The Lie algebra $\mathfrak{g}$ of $G$ is then a certain vector space of $q \times q$ matrices (not necessarily invertible), which is closed under the bracket operation: $u, v \in \mathfrak{g}$ implies $[u, v] \in \mathfrak{g}$ (where $[u, v] = uv - vu$).

This Lie algebra $\mathfrak{g}$ admits a Cartan decomposition $\mathfrak{g} = \mathfrak{k} + \mathfrak{p}$. This is a direct sum of vector spaces where $\mathfrak{k}$ is the Lie algebra of $K$ and $\mathfrak{p}$ is a subspace of $\mathfrak{g}$, which can be identified with the tangent space to $M$ at $x_o$.

A subspace $\mathfrak{a}$ of $\mathfrak{p}$ is called abelian if $[u, v] = 0$ for any $u, v \in \mathfrak{a}$. If $\mathfrak{a}$ is such an abelian subspace, which has maximal dimension, then each $v \in \mathfrak{p}$ can be put under the form $v = kak^{-1}$ for some $k \in K$ and $a \in \mathfrak{a}$ — note that this is just a matrix product. In fact, it should be thought of as a "diagonalisation" of $v$.

Formulas (51) and (52) from Appendix B remain true in general. The radial curvature operator is given by

$$R_s(u) = [s, [s, u]] \qquad s, u \in T_{x_o} M \simeq \mathfrak{p} \tag{54}$$

where $s, u$ belong to $T_{x_o} M$ (the tangent space to $M$ at $x_o$), which is identified with the subspace $\mathfrak{p}$ of $\mathfrak{g}$. It is important to understand that (54) should only be applied after $s, u$ have been recast as elements of $\mathfrak{p}$.

With this in mind, if $s = k\varsigma k^{-1}$, with $k \in K$ and $\varsigma \in \mathfrak{a}$, the eigenvalues of the radial curvature operator $R_s$ are the following: 0 with multiplicity $N$ (known as the rank of $M$), and $-(\kappa(\varsigma))^2$ with respective multiplicities $m_\kappa$, where each $\kappa : \mathfrak{a} \to \mathbb{R}$ is a linear function, knows as a positive restricted root of the Lie algebra $\mathfrak{g}$ with respect to the subspace $\mathfrak{a}$.

As in Appendix B, the eigenvalues of the radial curvature operator play the role of principal sectional curvatures. The sectional curvatures of $M$ are therefore included between the minimum and maximum of these eigenvalues. It follows that $M$ has negative sectional curvatures, and that these are bounded below by

$$-\kappa^2 = \min -(\kappa(\varsigma))^2 \tag{55}$$

where the minimum is taken over all positive restricted roots $\kappa$ and all $s \in S_{x_o} M$, where $s = k\varsigma k^{-1}$.

The eigenvalues of the radial curvature operator also allow one to directly write down the volume density $\det A(r, s)$. Each eigenvalue equal to 0 contributes a factor $r$, while each eigenvalue $-(\kappa(\varsigma))^2$ contributes a factor $\sinh(\kappa(\varsigma)r)/\kappa(\varsigma)$. Just as in Appendix B (or also in Appendix C), this yields

$$\det A(r, s) = r^{N-1} \prod_\kappa (\sinh(\kappa(s)r)/\kappa(s))^{m_\lambda} \tag{56}$$

where the product is over all positive restricted roots. Note here that $\kappa(\varsigma)$ has been written as $\kappa(s)$, by analogy with (22). There is no ambiguity is using this notation, since one can prove that the product in (56) only depends on $s$ and not on the particular decomposition $s = k\varsigma k^{-1}$.

Finally, for the acceptance probability $\Pi$ in (15), Formulas (16) and (56) should be used in computing $Z$, while Formula (17) remains the same for $Z_\kappa$. The issue of obtaining expressions similar to the one in (28) (in other words, determinantal or Pfaffian expressions for $Z$), in the general setting described by (16) and (56), is the subject of ongoing research(Said & Mostajeran, 2023).

To make the above general treatment of negatively-curved symmetric spaces more concrete, it is helpful to consider three basic examples. In these examples, $M = M_\beta$ is the space of $N \times N$ covariance matrices, whose entries are real numbers, complex numbers, or quaternions, according to $\beta = 1, 2$, or $4$. The group $G$ of isometries is the group of transformations $g : M_\beta \to M_\beta$ of the form $g(x) = gxg^\dagger$, where $g$ is an $N \times N$ invertible (real, complex, or quaternion) matrix and $\dagger$ denotes the conjugate-transpose (Said, 2021).

Therefore, $G$ may be identified with the matrix Lie group $GL(N, \mathbb{K}_\beta)$ of invertible matrices $g$ with entries from $\mathbb{K}_\beta = \mathbb{R}, \mathbb{C}$ or $\mathbb{H}$. The Lie algebra $\mathfrak{g}$ of $G$ is typically denoted $\mathfrak{gl}(N, \mathbb{K}_\beta)$, and is just the vector space of $N \times N$ matrices $v$ (not necessarily invertible) with entries from $\mathbb{K}_\beta$.

The Cartan decomposition is $\mathfrak{g} = \mathfrak{k} + \mathfrak{p}$, where $\mathfrak{k}$ is the subspace of matrices $v \in \mathfrak{g}$ such that $v + v^\dagger = 0$, and $\mathfrak{p}$ is the subspace of matrices $v \in \mathfrak{g}$ such that $v - v^\dagger = 0$. Therefore, matrices in $\mathfrak{k}$ are skew-Hermitian and those in $\mathfrak{p}$ are Hermitian.

The subspace $\mathfrak{k}$ of $\mathfrak{g}$ is the Lie algbera of the subgroup $K$ of $G$, which is made up of $g \in G$ such that $gg^\dagger = \mathrm{id}$ (Said, 2021). These are exactly those $g \in G$ for which $g(x_o) = x_o$ when $x_o$ is taken to be id (the $N \times N$ identity matrix). On the other hand, the subspace $\mathfrak{p}$ of $\mathfrak{g}$ is isomorphic to the tangent space to $M_\beta$ at $x_o = \mathrm{id}$. It is usual to identify $u$ in this tangent space with $\tilde{u} = \frac{1}{2}u$ in $\mathfrak{p}$.

A maximal abelian subspace $\mathfrak{a}$ of $\mathfrak{p}$ is (for example) the space of $N \times N$ real diagonal matrices (these are real even when $\mathbb{K}_\beta = \mathbb{C}$ or $\mathbb{H}$). This is clearly abelian because $[u, v] = 0$ whenever $u$ and $v$ are diagonal matrices. Then, any $v \in \mathfrak{p}$ can be written $v = kak^{-1}$ where $k \in K$ and $a \in \mathfrak{a}$ — this is just a general statement of the fact that any Hermitian matrix is unitarily diagonalisable.

The unit sphere $S_{x_o} M_\beta$ is the set of Hermitian matrices $s$ with $\mathrm{tr}(s^2) = 1$. The radial curvature operator $R_s$ is given by (54). Here, one should be careful to replace $s$ by $\tilde{s} = \frac{1}{2}s$ and $u$ by $\tilde{u} = \frac{1}{2}u$ into (54). Then,

$$R_s(u) = \frac{1}{4}[s, [s, u]] \tag{57}$$

This has a factor $\frac{1}{4}$ instead of $\frac{1}{8}$ because after computing the right-hand side of (54), one has to multiply by $2$ in order to return from $\mathfrak{p}$ to the tangent space at $x_o$ of $M_\beta$ (*i.e.* from $\tilde{v} = \frac{1}{2}v$ to $v$, where $v$ denotes $R_s(u)$).

If $s = k\varsigma k^{-1}$ where $k \in K$ and $\varsigma \in \mathfrak{a}$, so $\varsigma$ is a real diagonal matrix, then the eigenvalues of $R_s$ are the following: $0$ with multiplicity $N$, and $-(\kappa_{ij}(\varsigma))^2$ for $1 \leq i < j \leq N$, each one with multiplicity $\beta$, where $\kappa_{ij}(\varsigma) = (\varsigma_i - \varsigma_j)/2$ when $\varsigma = \mathrm{diag}(\varsigma_i; i = 1, \ldots, N)$. For the three examples at hand ($M = M_\beta$), these $\kappa_{ij}$ (reminiscent of (23)) are the positive restricted roots.

From (55), it is possible to find $\kappa = 1/\sqrt{2}$, while (56) gives the volume density

$$\det A(r, s) = r^{N-1} \prod_{i<j} (\sinh(\kappa_{ij}(s)r)/\kappa_{ij}(s))^\beta \tag{58}$$

which reduces to (22) when $\beta = 1$.

The three examples just considered are rather immediate generalisations of the space of real covariance matrices from Section 5. However, an extensive variety of interesting matrix spaces can be understood using symmetric space techniques which were reviewed in the present appendix (see (Edelman & Jeong, 2023)). Using the information about positive restricted roots (for example in (Said et al., 2018)), Formula (56) can be written down for other spaces of covariance matrices, such as spaces of Toeplitz or block-Toeplitz covariance matrices, and this directly leads to the adequate implementation of CURS.

# E   Further numerical experiments

The numerical experiments reported in Section 5 were limited to the space $M$ of $N \times N$ real covariance matrices, and to Gaussian distributions with the specific exponent $\alpha = 2$ in (19). Here, further experiments are reported which explore what happens beyond these limitations.

## E.1   Complement to Section 5

Keeping the same $M$ as in Section 5 (the space of $N \times N$ real covariance matrices), consider the effect of changing the value of $\alpha$ in (19). The following Tables 4 and 5 give the acceptance probability $\hat{\Pi}_s$ obtained by using sharp CURS (Paragraph 5.2)) when $N = 4$ and the exponent $\alpha$ is either $\alpha = 4$ or $\alpha = 1.5$. Each table also contains the values of the empirical variance $\hat{\delta}$. This is the empirical version (that is the estimate) of the variance parameter $\delta$ in (30), computed directly from the samples generated by CURS.

As in Section 5, standard deviations are not provided, since they are too small to offer additional information. In comparison to Table 1, which dealt with $\alpha = 2$, Table 4 shows that increasing $\alpha$ from 2 to 4 significantly improves the acceptance probability $\hat{\Pi}_s$. This should come as no surprise, since the density (19) becomes more concentrated near its mode $x_o$ as $\alpha$ increases. On the other hand, Table 5 has much weaker acceptance probabilities than Table 1. As evidenced by the values of $\hat{\delta}$, this is because the density (19) with $\alpha = 1.5$ is very widely scattered away from $x_o$, even for apparently small values of $\sigma$. When the acceptance probability nearly drops to zero, in the last column of Table 5, it is not possible to evaluate $\hat{\delta}$.

## E.2   Complex covariance matrices

Recall, from the previous appendix, the spaces $M_\beta$ of $N \times N$ covariance matrices with real, complex or quaternion entries, according to $\beta = 1, 2$ or $4$. For these spaces, the volume density $\det A(r, s)$ given by (58). Since (58) is a straightforward generalisation of (22) from Section 5, the application of CURS (in either its general or sharp forms) to the spaces $M_\beta$ closely mirrors the development in Section 5.

Consider here the case $\beta = 2$ of $N \times N$ complex covariance matrices. The overall steps 1 to 8 of Algorithm 1 remain as in Paragraph 5.1, except for the following modifications: while $\kappa = 1/\sqrt{2}$ as before, the dimension is now $d = N^2$. For Step 1, $s$ is an $N \times N$ Hermitian (instead of symmetric) matrix. First, $s$ should be generated from a Gaussian unitary (instead of orthogonal) ensemble. Then, it should be replaced with $s/\|s\|$ where $\|s\|^2 = \mathrm{tr}(s^2)$ — the Gaussian unitary ensemble has already been introduced in Section 6.

Table 4: ($N = 4$ and $\alpha = 4$) Empirical acceptance probabilities for sharp CURS ($\hat{\Pi}_s$)

| $\sigma$ | 0.2 | 0.4 | 0.6 | 0.8 | 1.0 | 1.2 | 1.4 |
|---|---|---|---|---|---|---|---|
| $\hat{\delta}$ | 0.4284 | 0.8621 | 1.3024 | 1.7475 | 2.1974 | 2.6564 | 3.1125 |
| $\hat{\Pi}_s$ | 0.8611 | 0.7405 | 0.6364 | 0.5453 | 0.4680 | 0.4016 | 0.3430 |

Table 5: ($N = 4$ and $\alpha = 1.5$) Empirical acceptance probabilities for sharp CURS ($\hat{\Pi}_s$)

| $\sigma$ | 0.1 | 0.2 | 0.3 | 0.4 | 0.5 |
|---|---|---|---|---|---|
| $\hat{\delta}$ | 0.3342 | 2.0039 | 6.2811 | 15.9730 | |
| $\hat{\Pi}_s$ | 0.8888 | 0.4833 | 0.0904 | 0.0017 | 0.0000 |

For Step 4, the rejection condition (27) should be replaced with

$$U > (\kappa r / \sinh(\kappa r))^{N-1} \prod_{i<j} \left[ \frac{(\sinh(\kappa_{ij}(s)r)/\kappa_{ij}(s))}{(\sinh(\kappa r)/\kappa)} \right]^{\beta} \tag{59}$$

where $\beta = 2$ in the present (*i.e.* complex) case. This condition corresponds to the general version of CURS (Paragraph 5.1). To obtain the sharp version (Paragraph 5.2), it is enough to omit the factor $(\kappa r / \sinh(\kappa r))^{N-1}$ before the product.

In the case of Gaussian distributions, where $\alpha = 2$ in (19), it is now possible to produce results similar to those of Figure 2. Specifically, Figure 5 shows the plot of the theoretical acceptance probability $\Pi$ as a function of the variance $\delta$ given by (30), overlaid with individual values of the empirical acceptance probability $\hat{\Pi}$, which are marked by dot points. The theoretical acceptance probability $\Pi$ is found from the general formula (15), with $Z_\kappa$ given by the same (29) as in Paragraph 5.1 (here, of course, with the correct $d = N^2$), and with $Z$ given by the new formula (instead of (28))

$$Z(\sigma) = \left( \frac{\pi}{2} \right)^{\frac{N^2}{2}} \sigma^N \prod_{j=1}^{N-1} \frac{1}{j!} \left( e^{j\sigma^2} - 1 \right)^{N-j} \tag{60}$$

which holds for any $N$ (Said, 2021; Said et al., 2023). The theoretical and empirical acceptance probabilities are seen to agree quite closely. In order to improve upon these acceptance probabilities, it is possible to resort to sharp CURS. The following table 6 reports the empirical acceptance probabilities from CURS ($\hat{\Pi}$) and sharp CURS ($\hat{\Pi}_s$), obtained for certain values of $\delta$ when the matrix size is $3 \times 3$ (these are the values of $\delta$ for the dot points in Figure 5b). The improvement achieved by sharp CURS is clear from this table.

Table 6: (Complex matrices of size $3 \times 3$) Empirical acceptance probabilities $\hat{\Pi}$ and $\hat{\Pi}_s$

| $\delta$ | 0.3665 | 1.5465 | 3.8048 | 7.6554 | 13.9540 |
|---|---|---|---|---|---|
| $\hat{\Pi}$ | 0.8484 | 0.4914 | 0.1614 | 0.0220 | 0.0009 |
| $\hat{\Pi}_s$ | 0.9014 | 0.6347 | 0.3023 | 0.0756 | 0.0082 |

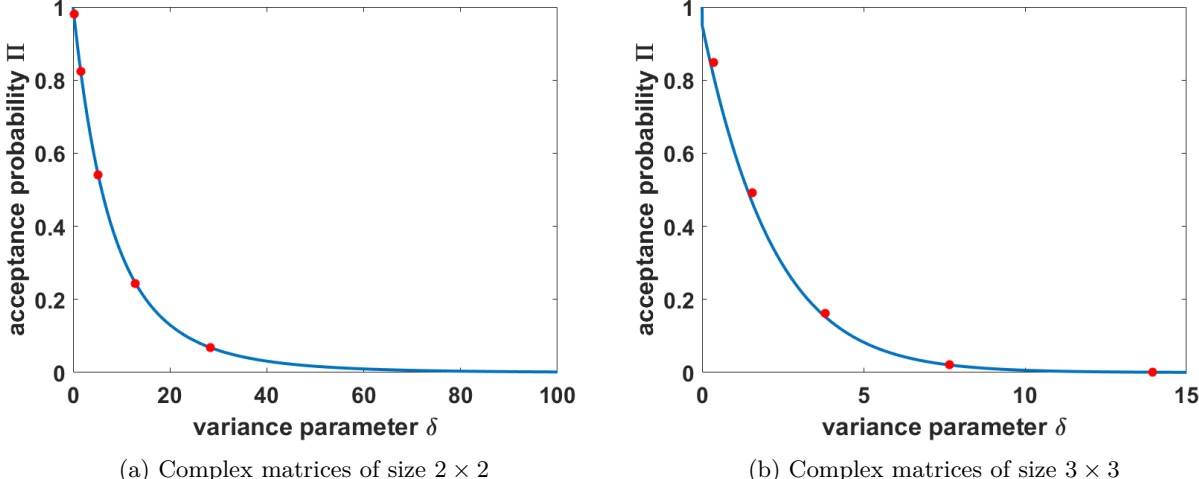

(a) Complex matrices of size $2 \times 2$      (b) Complex matrices of size $3 \times 3$

Figure 5: Theoretical (solid line) and empirical (dot points) acceptance probabilities, plotted against $\delta$

### E.3 Runtime experiments

Recall from (15) that the expected number of iterations that CURS requires to produce one new sample is $T = 1/\Pi$. The question of how much time, say how many seconds, CURS requires to produce a sample then boils down to how long a single iteration takes to run.

A general idea of the complexity of a single iteration of CURS can be found from Algorithm 1. By far the most difficult step is Step 8, but this is only evaluated when a sample is accepted. On average, Step 8 is evaluated once in every $T$ iterations.

All the remaining steps (Steps 1 to 7) have a complexity which is at most linear in the dimension $d$ of the manifold $M$. In fact, the complexity of Steps 2, 3, 5 and 6 does not depend on $d$ at all. Steps 1 and 4 have linear complexity, as now explained.

In order to carry out Step 1, as discussed right after Algorithm 1, one must generate $d$ independent Gaussian random variables, the components of the vector $s$ in the tangent space to $M$ at $x_o$, and then compute $s/\|s\|$. Since $\|s\|^2$ is the sum of squared components, the overall complexity is clearly linear in $d$.

For Step 4, it is assumed that $\det A(r, s)$ is known in closed form. Then, formulas like (22) or the more general (56) show that evaluating $\det A(r, s)$ requires a number of multiplications that grows linearly with $d$.

The curse of dimensionality entails that the expected number of iterations for one sample, that is $T$, will also grow with the dimension $d$. In practice, the main difficulty in using CURS comes from $T$ being too large, rather than from the complexity of individual iterations.

A few numerical observations about this situation. Let $M$ be the space of $N \times N$ real covariance matrices. When $N = 2$ and $\alpha = 2$ in (19), the average time required to obtain 1000 samples from the target density, using sharp CURS, increases from about 16 seconds to about 30 seconds, as $\sigma$ ranges over the interval $[0.1, 2]$. The average is taken over 20 runs of CURS (so 20000 samples are generated for each value of $\sigma$) on a standard desktop computer.

The same average time required to obtain 1000 samples, when $N = 4$ or $6$, is shown in Figure 6 below. For $N = 4$, this amounts to about 40 minutes if $\sigma = 1$. For $N = 6$, it is already near 6 hours when $\sigma = 0.5$.

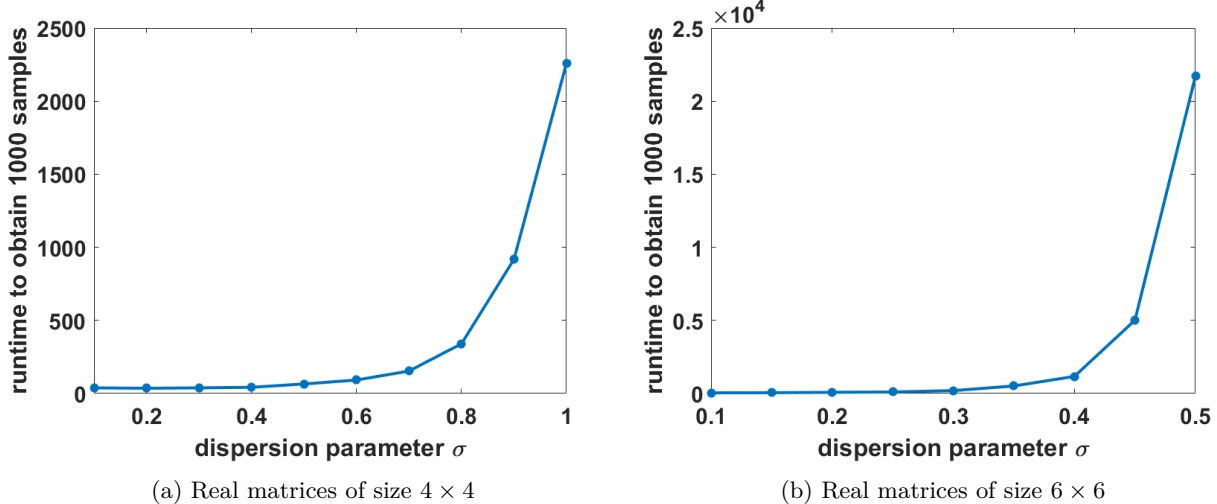

(a) Real matrices of size $4 \times 4$      (b) Real matrices of size $6 \times 6$

Figure 6: Runtime required to obtain 1000 samples (in seconds) plotted against $\sigma$

