# OpenReview forum: "CURS: An exact method for sampling on Riemannian manifolds"
_TMLR — Accepted by TMLR_

### Review · Reviewer_Ew29 · 2026-02-17

**Summary Of Contributions:**

The contribution of this paper can be summarized as providing an exact sampling method for distance-dependent densities by leveraging lower curvature bounds and volume comparison theorems.
Specifically, for the class of distributions $p(x) \propto f(d(x_0, x))$, the author addresses the fundamental challenge where volume density in geodesic polar coordinates is non-separable in terms of direction and radius. The paper presents a clear algorithm that constructs a uniform upper bound using Bishop's volume comparison inequality, defines an independent proposal distribution based on this bound, and corrects it via rejection sampling. The core of this work lies in directly translating the purely geometric assumption of a lower curvature bound into the construction of a computable proposal distribution.

Furthermore, this method is truly exact rather than MCMC-based, meaning it does not rely on burn-in or asymptotic convergence. While the rejection rate determines the essential complexity, providing a framework to theoretically evaluate the acceptance probability as a ratio of normalization constants is a quantitative contribution that goes beyond a simple methodological proposal. In particular, the derivation of concrete formulas for negative-curvature symmetric spaces (such as the manifold of SPD covariance matrices) where the volume density has a closed form—along with the introduction of a "sharp" version that improves the acceptance rate by accounting for directions with zero principal curvature—demonstrates a refined exploitation of curvature structures.

On the other hand, this is not a theory for general densities on arbitrary manifolds, but a method specialized for the clearly restricted class of distance-dependent densities. The author(s) also acknowledge that the rejection rate deteriorates rapidly in high dimensions. Therefore, the essence of the contribution lies not in broad generality, but in the elegant construction that directly transforms classical geometric results into a sampling algorithm and its exactness and theoretical analysis of the acceptance rate.

**Audience:**

Yes

**Audience Explanation:**

Sampling methods are an essential building block frequently required in both the theory and application of machine learning, and there is a steady demand for simple, exact methods tailored to specific classes of distributions.

**Broader Impact Concerns:**

Not applicable.

**Claims And Evidence:**

Yes

**Claims Explanation:**

Figures 1 and 2 clearly show that the theoretical rejection rate matches empirical one.

**Requested Changes:**

A similar study exists titled "Riemannian Proximal Sampler for High-accuracySampling on Manifolds."
https://arxiv.org/abs/2502.07265

Both works share the following framework:
- Sampling from probability densities on Riemannian manifolds.
- Explicitly utilizing geometric quantities (distance, curvature, heat kernels).

More specifically, while the CURS proposed in this paper takes the approach of upper-bounding the volume density via Bishop's comparison theorem, the Riemannian Proximal Sampler defines conditional distributions using the heat kernel. Although their underlying philosophies differ, both share the core ideology of "sampling that preserves geometric structure." Regarding the class of target distributions, CURS is limited to distance-dependent densities, whereas the theoretical scope of the Proximal Sampler is significantly broader. Furthermore, the latter theoretically controls dimension dependence. While CURS (an exact sampler) and the Proximal sampler appear to be complementary, it would be beneficial to clarify their respective Pros and Cons.
Additionally, a comparative experiment on target distributions where both are applicable, such as Gaussian distributions on the SPD manifold, would be desirable (though not strictly mandatory).

The format should adhere to the TMLR template and instructions. Specifically, table names and captions are required to be placed above the tables. Also, the appendix should be placed after the references.

Labels for the horizontal and vertical axes should be provided for each plot in Figures 1 and 2.

---

> ### Author Response · Authors · 2026-04-02
> **Initial response**
>
> Thank you for your comments and suggestions.
>
> We have prepared a new version of our submission, which fixes the formatting issues which you mentioned (for tables, figures,
> and appendices).
>
> This new version also discusses the Riemannian proximal sampler, from the reference which you have provided. After careful
> examination of this reference and of the accompanying code, we have arrived at two conclusions :
>
> ** high-accuracy sampling can produce nearly-exact independent samples, and is not restricted to distance-dependent densities. This is quite interesting as it overcomes one of the fundamental limitations of CURS.
>
> ** for distance-dependent densities, it seems CURS is preferable to high-accuracy sampling, due to its lower computational cost. On the other hand, when dealing with more general densities, CURS can be very useful as a component or sub-part of methods such as the Riemannian proximal sampler.
>
> With regard to the influence of dimension, we feel both methods are more or less similarly affected by the curse of dimensionality. However, this issue certainly requires a more detailed and fair comparison.
>
> Finally, we have performed a new numerical experiment, involving the same density function as in Appendix A.3 of the reference which you have provided.
>
> EDIT : we have now posted a revised version. Important changes are highlighted in red.

---

### Review · Reviewer_vmoy · 2026-03-02

**Summary Of Contributions:**

This manuscript proposes a new algorithm for rejection sampling from (generalized) Gaussian distributions on Riemannian manifold. The algorithm is called CURS or curvature-based rejection sampling. The authors start by describing the general idea behind CURS as well as its geometric interpretation. They then extend specialize CURS to sample a generalized Gaussian density on the space of NxN real covariance matrices and introduce a modification called sharp CURS to improve the acceptance probabilities of the algorithm. A discussion of how to deal with the cut locus is provided in the particular case of the group of NxN unitary matrices.

Strengths:

(1) In contrast to MCMC-based techniques, the proposed algorithm generates exact samples from the underlying distribution.
(2) The algorithm appears to be fast and efficient in low dimensions.
(3) The algorithm is rather general and can be applied in the context of many Riemannian manifolds.
(4) The authors provide a geometric interpretation of the algorithm, which improves conceptual understanding.
(5) The manuscript is written very well.

Weaknesses:

(1) While this is clearly acknowledged in the manuscript, the algorithm is heavily affected by the dimensionality of the underlying manifold, and is thus only applicable in fairly low-dimensional settings.
(2) Simulations and examples are limited.
(3) Computational cost of the algorithm is only vaguely stated and is not analyzed in detail.

**Audience:**

Yes

**Audience Explanation:**

Sampling from distributions on Riemannian manifolds is an important problem in machine learning.

**Claims And Evidence:**

Yes

**Claims Explanation:**

The technical content of this manuscript appears to be accurate. The authors clearly state limitations of the proposed approach, e.g., due to the curse of dimensionality. Some simulations are provided to assess correctness via comparison of theoretical and empirical acceptance probabilities.

**Requested Changes:**

(1) The authors should expand the scope of their simulations by considering other Riemannian manifolds in addition to the space of real covariance matrices. While I understand that there is a 12 page limit, broader experiments could be included as part of an appendix (I am not suggesting that any of the presented material should be replaced by additional simulations).

(2) The authors could expand their discussion of applications where sampling from Riemannian Gaussian distributions is important. This aspect is briefly mentioned in the manuscript with a reference at the beginning of Section 5. Expanding on this would broaden the appeal of this manuscript to a more general TMLR audience.

(3) The authors should report the computational cost of the CURS algorithm for a few cases. There is a brief discussion of this in the context of 4x4 covariance matrices.

---

> ### Author Response · Authors · 2026-04-02
> **Initial response**
>
> Thank you very much for your comments and suggestions.
>
> We have prepared a new version of our submission, which hopefully deals with the three issues you have mentioned :
>
> (1) see our new appendix E which contains new simulations dealing with :
>
> a) the situation of Section 5 when the exponent alpha is different from 2 ;
>
> b) application of CURS to sampling on the space of complex (rather than real) covariance matrices ;
>
> c) a discussion of the computational cost of CURS, in addition to experiments which measure the time required to generate a given number of samples
>
> (2) the introduction has two new paragraphs, one of them dedicated to recalling several machine-learning problems which involve the application of sampling on manifolds.
>
> (3) hopefully, the new Appendix E.3 will provide an adequate initial discussion of computational complexity and runtime. The experiments in this appendix can easily be repeated for other spaces or densities, but we do not expect any qualitative difference to appear.
>
> EDIT : we have now posted a revised version. Important changes are highlighted in red.

---

### Review · Reviewer_AFou · 2026-05-06

**Summary Of Contributions:**

This paper proposes a method for sampling from probability densities on Riemannian manifolds of the form $p(x) \propto f(d(x_0,x))$. They upper bound the volume form of the manifold in equation 8, which holds when the curvature of the manifold is bounded below. This allows them to introduce a rejection sampling scheme and a proposal distribution that is easy to sample because it factorizes the components corresponding to $r$ and $s$. This approach holds only when geodesic spherical coordinates cover the entire manifold. In Section 6 they generalize the proposed algorithm to general manifolds. They validate the proposed sampling scheme on a manifold of SPD covariance matrices.

**Audience:**

Yes

**Audience Explanation:**

Yes. The paper addresses an interesting and nontrivial problem: exact sampling from probability distributions defined intrinsically on Riemannian manifolds. This is relevant to parts of the TMLR audience working on geometric machine learning, manifold-valued latent-variable models, Riemannian statistics, Bayesian inference on manifolds, and non-Euclidean generative modelling.

**Broader Impact Concerns:**

No broader impact concerns.

**Claims And Evidence:**

Yes

**Claims Explanation:**

Yes, they are, but with some qualifications. The high-level mathematical idea is simple but elegant. However, I am unsure about some of the details that I would like to be clarified.

1. In Algorithm 1, line 6 says reject $(r,s)$ and return to step 2. But we sample $s$ step 1. Is this a typo? Shouldn't we sample both a new $r$ and $s$ and not only a new $r$? If $s$ is kept fixed until an $r$ is accepted, the marginal distribution of $s$ is uniformly distributed, which is not what we want in general.
2. I am not sure I find the empirical evidence in section 5.1 to be convincing. They comapre the acceptance probabilities of CURS with geometric Langevin algorithm. But I dont think that is suffiecient to validate that the accepted samples have the correct distribution.
3. The general algorithm in Section 6 is a bit unclear. The authors should clarify whether they are proposing a general method for arbitrary manifolds with cut loci, or only giving examples where the $c-$function is known. if a closed expression for it is needed then the practical applicability is limited.

**Requested Changes:**

1. Correct line 6 in Algorithm 1.
2. Add experiments to validate that the accepted samples follow the target distribution.
3. State the assumptions for exactness more clearly. Assumptions such as bounds on the curvature, finite normalizing constant for the proposal, known cut-locus truncation, etc., are stated verbally and scattered throughout the text. A systematic presentation fot ehm would benefit readability
4. Improve the writing in section 6 mainly by clarifying the role of the cut-locus.

---

> ### Author Response · Authors · 2026-05-21
> **Initial response**
>
> Thank you very much for your detailed reading of our submission. With regard to the changes you have requsted, we have made the following modifications (the numbering is the same as in your message) :
>
> 1.	thank you for picking up this mistake, which is now corrected.
>
> 2.	please have a look at the new Paragraph 5.3, which briefly discusses the question of « checking that the samples follow the correct distribution ». We honestly feel this is the most detailed discussion we can provide within the scope of our present work.
>
> 3.	the fundamental assumptions have been separated from the main flow of the text and labeled Assumptions A (regularity of spherical coordinates), B (lower bound on sectional curvatures), C (explicitly known volume density), D (explicitly known c-function). Moreover, this labeling is used for reference in certain other places of the text when the assumptions are discussed.
>
> 4. we have modified the opening part of Section 6, emphasising two things : first, it is a requirement (assumption) that the c-function should be known explicitly. Second, this section illustrates the general idea through a specific example, without attempting a complete formulation.
>
> In the revised version of our submission, important changes are highlighted in red.

---

### Decision · Action_Editor_ndyA · 2026-06-01

**Recommendation:** Accept as is

**Additional Comments:**

Please update the PDF to avoid highlighting changes in red, and include author names, acknowledgements, and any other changes that align with the reviewer's feedback.

**Audience:**

Yes

**Audience Explanation:**

Sampling on manifolds is relevant to practitioners of generative models leveraging geometric inductive biases to shape latent spaces, shape statistics, to theoretical works to understand invariances in deep learning, and more.

**Claims And Evidence:**

Yes

**Claims Explanation:**

The paper is clear in its contributions, which are both backed up theoretically and through empirical tests.

---

> ### Author Response · Authors · 2026-06-02
> **Thank you**
>
> Thank you very much for this great news
>
> We will do our best to submit final material as soon as possible,
> and certainly before the required deadline
>
> EDIT : we have now uploaded the camera ready version.
> Thank you once more.